# Time-Dependent VAE for Building Latent Representations from Visual Neural Activity with Complex Dynamics

## Abstract

Seeking high-quality representations with latent variable models (LVMs) to reveal the intrinsic correlation between neural activity and behavior or sensory stimuli has attracted much interest. Most work has focused on analyzing motor neural activity that controls clear behavioral traces and has modeled neural temporal relationships in a way that does not conform to natural reality. For studies of visual brain regions, naturalistic visual stimuli are high-dimensional and time-dependent, making neural activity exhibit intricate dynamics. To cope with such conditions, we propose Time-Dependent Split VAE (TiDeSPL-VAE), a sequential LVM that decomposes visual neural activity into two latent representations while considering time dependence. We specify content latent representations corresponding to the component of neural activity driven by the current visual stimulus, and style latent representations corresponding to the neural dynamics influenced by the organism's internal state. To progressively generate the two latent representations over time, we introduce state factors to construct conditional distributions with time dependence and apply self-supervised contrastive learning to shape them. By this means, TiDeSPL-VAE can effectively analyze complex visual neural activity and model temporal relationships in a natural way. We compare our model with alternative approaches on synthetic data and neural data from the mouse visual cortex. The results show that our model not only yields the best decoding performance on naturalistic scenes/movies but also extracts explicit neural dynamics, demonstrating that it builds latent representations more relevant to visual stimuli.

## 1 Introduction

With the rapid development of neural recording technologies, researchers are now able to simultaneously record the spiking activity of large populations of neurons, providing new avenues for exploring the brain (Urai et al., 2022). For analyzing these high-dimensional data, an important scientific problem is how to account for the intrinsic correlation between neural activity and behavioral patterns or sensory stimuli. As an influential approach, latent variable models (LVMs) construct low-dimensional latent representations bridging to behavior or stimuli and explain neural activity well (Saxena & Cunningham, 2019; Bahg et al., 2020; Vyas et al., 2020; Jazayeri & Ostojic, 2021; Langdon et al., 2023). Recently, advanced deep learning algorithms allowed LVMs to extract high-quality representations from neural activity without knowledge of experimental labels (Wu et al., 2017; Pandarinath et al., 2018; Glaser et al., 2020; Liu et al., 2021), or to incorporate behavioral information into models to constrain the shaping of latent variables (Mante et al., 2013; Hurwitz et al., 2021; Sani et al., 2021; Singh Alvarado et al., 2021; Ahmadipour et al., 2024). These approaches have made various contributions to the analysis of neural activity, such as predicting held neural responses (Gao et al., 2016; Pandarinath et al., 2018; Kapoor et al., 2024), decoding related motion patterns or simple visual scenes (Liu et al., 2021; Schneider et al., 2023), and constructing interpretable latent structures (Zhou & Wei, 2020; Aoi et al., 2020).

However, most studies have dealt with neural data recorded from motor brain areas under specific controlled behavioral settings (Churchland et al., 2012; Pandarinath et al., 2018; Zhou & Wei, 2020; Liu et al., 2021; Pei et al., 2021), such as pre-planned reaching movements (Dyer et al., 2017). There is little work using LVMs to analyze neural data from visual brain regions (Gao et al., 2016; Zhao

& Park, 2017; Schneider et al., 2023), even though how the visual system encodes input to recognize objects is a primary topic (DiCarlo et al., 2012), and decoding visual neural activity to visual stimuli is a challenging research highlight in the neuroscience community (Kay et al., 2008; Wen et al., 2018; Du et al., 2023). Furthermore, existing LVMs treat temporal relationships unnaturally (Pandarinath et al., 2018; Schneider et al., 2023) or even don't consider time dependence (Zhou & Wei, 2020; Palmerston & Chan, 2021). Given that visual neural activity has strict antecedent time dependence, these models may struggle to build high-quality latent representations.

In this work, we propose Time-Dependent Split VAE (TiDeSPL-VAE), a sequential LVM that builds two split latent representations with time dependence to better analyze visual neural activity. We adopt the practice of splitting latent variables into content and style representations (Liu et al., 2021). Content latent representations correspond to *the component of neural activity driven by the current visual stimulus*, while style latent representations correspond to *the neural dynamics influenced by the organism's internal state* (pupil position, signals relayed from other brain regions, the neurons' underlying currents, etc.). These latent variables are optimized by self-supervised contrastive learning. For comparison with outstanding alternatives, we evaluate our model on synthetic and mouse visual datasets. The results show that our model builds meaningful latent representations that are highly correlated with complex visual stimuli, providing new insights into the intrinsic relationship between neural activity and visual stimulation. Specifically, our main contributions are as follows.

- To construct highly time-dependent latent representations, we introduce state factors to accumulate and filter temporal information, allowing TiDeSPL-VAE to progressively compress neural activity along a chronological order in a natural way. Besides, we apply self-supervised contrastive learning to shape content latent variables.
- Through evaluation on synthetic datasets, we show that our model better recovers latent structure and is good at handling time-sequential data.
- Through evaluation on mouse visual datasets, we demonstrate that our model decodes neural activity to related natural scenes or natural movies well, showing the highest performance compared to alternative models. Furthermore, visualization of latent representations presents that our model captures explicit temporal structures of neural dynamics for different time scales.

## 2 RELATED WORK

With the advancement of deep learning, the application of cutting-edge learning algorithms and the innovative design of model structures have greatly promoted the development of LVMs in neuroscience. Some prominent works are summarized below.

**VAE-based LVMs for neural activity analysis**   Recently, VAE-based approaches have become a major avenue to discover latent variables underlying population neural activity, which better elucidates the mechanisms of neural representations. As a well-known model, latent factor analysis via dynamical systems (LFADS) used RNNs in a sequential VAE framework, extracting precise firing rate estimates and predicting observed behavior for single-trial data on motor cortical datasets (Pandarinath et al., 2018; Keshtkaran & Pandarinath, 2019; Keshtkaran et al., 2022). Through specific latent variable design, pi-VAE (Zhou & Wei, 2020) and Swap-VAE (Liu et al., 2021) built interpretable latent structures linked to motor behavioral patterns.

**LVMs for visual neural activity analysis**   Several studies have made an effort to extract latent manifolds from visual neural activity using LVMs. Although these studies cover various types of models, such as the Gaussian process model (Ecker et al., 2014; Gondur et al., 2024), linear dynamical system (Gao et al., 2016), autoencoder (Palmerston & Chan, 2021), and flow-based generative models (Bashiri et al., 2021), they are limited to simple visual stimuli and are used for the task of reconstructing neural responses. Recently, CEBRA, a self-supervised learning model, obtained consistent latent representations and made progress in decoding movies (Schneider et al., 2023).

## 3 TIME-DEPENDENT SPLIT VAE

**Basic notations**   Considering the neural activity of a population of neurons over a period of time, we define a sequence input as $\mathbf{x} = (\mathbf{x}_1, \mathbf{x}_2, \ldots, \mathbf{x}_T) \in \mathbb{R}^{T \times N}$, which represents spike counts of $N$

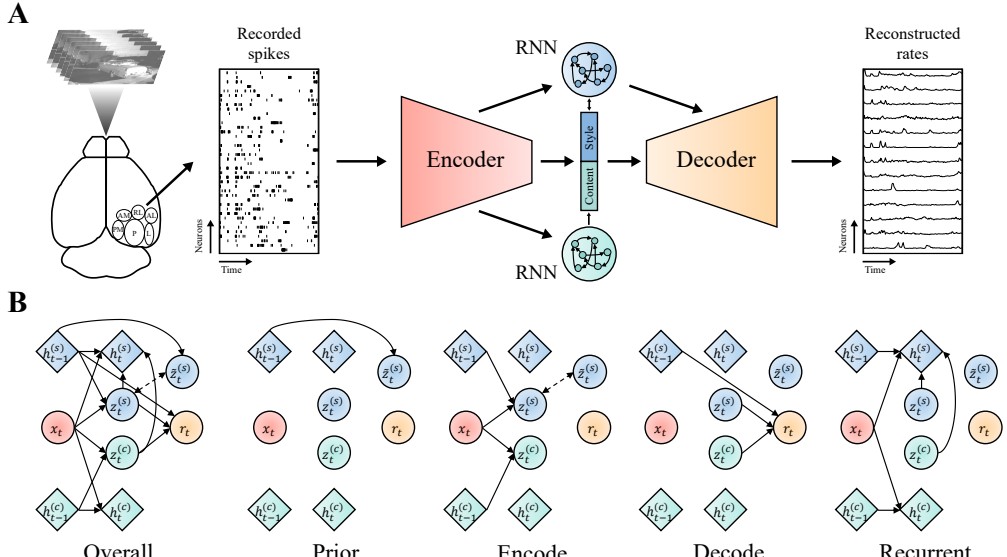

Figure 1: The method overview. **A**. The illustration of TiDeSPL-VAE for analyzing neural activity in the mouse visual cortex during passive viewing. The encoder extracts spatial features from sequential spike data. The latent variables are built conditionally on features of the encoder and RNNs' state factors to introduce time dependence. The decoder maps latent variables to reconstructed firing rates. Detailed network structures are given in Appendix A. **B**. The operations of each module in TiDeSPL-VAE (see details in Section 3.1).

neurons within $T$ time windows. The output $\mathbf{r} = (\mathbf{r}_1, \ldots, \mathbf{r}_T) \in \mathbb{R}^{T \times N}$ is an estimate of firing rates of the input. The low-dimensional latent representation of time point $t$ is denoted as $\mathbf{z}_t \in \mathbb{R}^M$.

## 3.1 MODEL ARCHITECTURE

Our goal is to compress neural activity into high-quality latent representations while modeling the temporal relationship. Therefore, we explicitly model the time dependence between latent variables across time steps based on a sequential VAE. In practice, we split latent variables into content and style latent representations ($\mathbf{z}_t = [\mathbf{z}_t^{(c)}, \mathbf{z}_t^{(s)}]$), corresponding to stimulus-driven and neural dynamical components, respectively. To construct time-dependent connections along a chronological order, we introduce the state factor $\mathbf{h}_t$ to sift and accumulate temporal information (Bayer & Osendorfer, 2015; Fabius & van Amersfoort, 2015; Chung et al., 2015) so that the latent variables and output of the current time step are conditioned on the input and state factors of the antecedent time steps. We name our model as Time-Dependent Split VAE (TiDeSPL-VAE; Figure 1A) and formulate the operations of each module (Figure 1B) below.

**Encode** Based on the above assumptions, content latent variables driven by the current stimulus are constructed as deterministic values, while style latent variables are constructed as random values from a parameterized distribution (approximate posterior) since there is a lot of intrinsic noise as well as variability in the neural dynamics. For time dependence, the latent variables are built on $\mathbf{x}_t$ and $\mathbf{h}_{t-1}$:

$$\mathbf{z}_t^{(c)} = f_{\text{enc}}^{(c)} \left( f_{\text{x}}(\mathbf{x}_t), \mathbf{h}_{t-1}^{(c)} \right), \tag{1}$$

$$\mathbf{z}_t^{(s)} \Big| \mathbf{x}_{1:t}, \mathbf{h}_{1:t-1}^{(s)} \sim \mathcal{N}(\boldsymbol{\mu}_{z,t}, \boldsymbol{\sigma}_{z,t}^2 \cdot \mathbf{I}), [\boldsymbol{\mu}_{z,t}, \boldsymbol{\sigma}_{z,t}] = f_{\text{enc}}^{(s)} \left( f_{\text{x}}(\mathbf{x}_t), \mathbf{h}_{t-1}^{(s)} \right), \tag{2}$$

where $f_x$, $f_{enc}^{(c)}$ and $f_{enc}^{(s)}$ are all parameter-learnable neural networks for extracting spatio-temporal features and building the latent variables. The similar functions of $f$ in the following text are also trainable neural networks.

**Prior**   Similar to the variational approximate posterior, the prior of $\mathbf{z}_t^{(s)}$ is conditioned on $\mathbf{h}_{t-1}$ for time dependence. The distribution is formulated as:

$$\tilde{\mathbf{z}}_t^{(s)}\Big|\mathbf{h}_{1:t-1}^{(s)} \sim \mathcal{N}(\tilde{\boldsymbol{\mu}}_{z,t}, \tilde{\boldsymbol{\sigma}}_{z,t}^2 \cdot \mathbf{I}), [\tilde{\boldsymbol{\mu}}_{z,t}, \tilde{\boldsymbol{\sigma}}_{z,t}] = f_{\text{prior}}^{(s)}\left(\mathbf{h}_{t-1}^{(s)}\right). \tag{3}$$

**Decode**   The decoder aims to reconstruct the neural activity input, which receives the full latent variables with style state factors as an auxiliary. Since the input is a sequence of spike counts, we denote the reconstructed responses as a parameterized Poisson distribution (Gao et al., 2016), i.e., the actual output of the decoder is spike firing rates:

$$\hat{\mathbf{x}}_t\Big|\mathbf{z}_{1:t}^{(c)}, \mathbf{z}_{1:t}^{(s)}, \mathbf{h}_{1:t-1}^{(s)} \sim \text{Poisson}(\mathbf{r}_t), \mathbf{r}_t = f_{\text{dec}}\left(\mathbf{z}_t^{(c)}, \mathbf{z}_t^{(s)}, \mathbf{h}_{t-1}^{(s)}\right). \tag{4}$$

**Recurrent**   The state factor is updated by recurrent neural networks, GRU (Cho et al., 2014). By selectively integrating and exploiting input and latent variables, the state factor is crucial for learning complex sequential dynamics. Importantly, since content state factors depend on dynamic changes in visual stimuli while stimulus-driven neural activity inevitably affects neural dynamics, $\mathbf{h}_t^{(c)}$ and $\mathbf{h}_t^{(s)}$ are updated differently:

$$\mathbf{h}_t^{(c)} = f_{\text{GRU}}^{(c)}\left(f_\text{x}(\mathbf{x}_t), \mathbf{h}_{t-1}^{(c)}\right), \tag{5}$$

$$\mathbf{h}_t^{(s)} = f_{\text{GRU}}^{(s)}\left(f_\text{x}(\mathbf{x}_t), \mathbf{z}_t^{(c)}, \mathbf{z}_t^{(s)}, \mathbf{h}_{t-1}^{(s)}\right). \tag{6}$$

### 3.2   Model Learning

As content latent variables correspond to the component of neural activity driven by visual stimuli, we use self-supervised contrastive learning to make them more relevant to visual stimuli. For a given sample $\mathbf{x} = (\mathbf{x}_1, \ldots, \mathbf{x}_T)$, we randomly select another sequence offset by several time steps as a positive sample, denoted $\mathbf{x}_{\text{pos}} = (\mathbf{x}_{1+\Delta}, \ldots, \mathbf{x}_{T+\Delta})$, where $\Delta$ can be positive or negative. In this work, the offset is always less than the length of the sequence to ensure that the positive sample pairs overlap and to enhance the time constraint. Then, a mini-batch of negative samples is randomly selected from the entire training set. The model is encouraged to bring the content latent variables of the positive pairs closer together and to push those of negative samples away.

As we introduce time dependence into our model and apply a time-dependent prior distribution to guide the parameterized distribution of style latent variables, we extend the evidence lower bound of VAE to a time-wise version and use the objective function for contrastive learning, which together form the loss function of TiDeSPL-VAE:

$$\mathcal{L} = \mathcal{L}_{\text{recons}} + \beta\mathcal{L}_{\text{regular}} + \gamma\mathcal{L}_{\text{contrast}}, \tag{7}$$

where $\beta$ and $\gamma$ are hyperparameters that serve to control the severity of the penalty for each loss term. The reconstruction loss $\mathcal{L}_{\text{recons}}$ is formulated as $\frac{1}{T}\sum_{t=1}^{T}\left[\mathcal{L}_{\text{P}}(\mathbf{x}_t, \mathbf{r}_t) + \mathcal{L}_{\text{P}}(\mathbf{x}_{\text{pos},t}, \mathbf{r}_{\text{pos},t})\right]$, where $\mathcal{L}_{\text{P}}$ is Poisson negative log likelihood loss. $\mathcal{L}_{\text{regular}}$ is the KL divergence to measure the difference between the prior and the approximate posterior of style latent variables, formulated as $\frac{1}{T}\sum_{t=1}^{T}\left[\text{D}_{\text{KL}}(\mathbf{z}_t^{(s)}\|\tilde{\mathbf{z}}_t^{(s)}) + \text{D}_{\text{KL}}(\mathbf{z}_{\text{pos},t}^{(s)}\|\tilde{\mathbf{z}}_{\text{pos},t}^{(s)})\right]$. Besides, we compute the L2 norm of the expectation and log-variance of the prior distribution as a regularization to avoid excessive fluctuations over time and to stabilize model training. We utilize NT-Xent loss (Chen et al., 2020) as $\mathcal{L}_{\text{contrast}}$. For this term, we do not apply the time-wise operation, but flatten the temporal and spatial dimensions of content latent variables for the loss computation. To enhance the effect of the positive sample, we adopt the practice of swapping content latent variables between the positive pairs while maintaining style latent variables (Liu et al., 2021). The swapped latent representations are then used to compute new reconstructed firing rates and an additional reconstruction loss. A detailed derivation of all formulas is given in Appendix B.

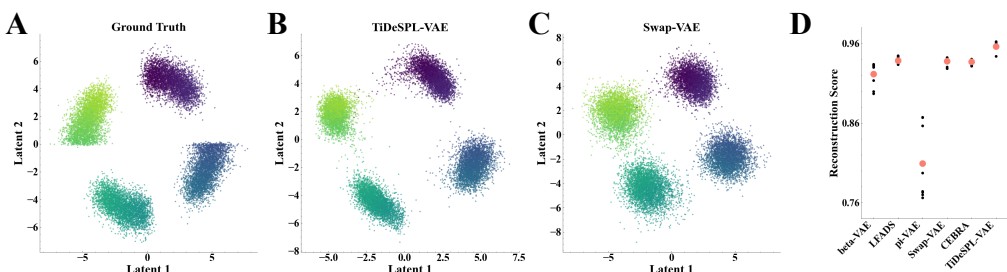

Figure 2: Results on the synthetic non-temporal dataset. **A**. The true latent variables. **B**. The regressed latent variables of our model. **C**. The results of an alternative model (see Appendix G for other alternatives). **D**. The reconstruction scores of all models. Each black dot represents an individual run, and the red dot represents the average score of ten runs.

## 4 EXPERIMENTS

### 4.1 EVALUATION AND ALTERNATIVE MODELS

In this work, we intend to construct latent representations that exhibit strong relevance to visual stimulation. According to this expectation, we evaluate our model from two aspects, similar to studies oriented to motor brain regions (Liu et al., 2021; Schneider et al., 2023). First, we quantify the performance in decoding visual stimuli using latent representations, which has long served as a research hotspot for unraveling the mechanisms of visual processing (Kay et al., 2008; Wen et al., 2018). Second, we assess the clarity of latent temporal structures extracted from neural dynamics.

For a comprehensive analysis, we compare TiDeSPL-VAE with several outstanding LVMs, including four generative models (an unsupervised: $\beta$-VAE (Higgins et al., 2017), a sequential: LFADS (Pandarinath et al., 2018), a supervised: pi-VAE (Zhou & Wei, 2020), and a self-supervised: Swap-VAE (Liu et al., 2021)) and a nonlinear encoding method with contrastive learning (CEBRA) (Schneider et al., 2023). Specifically, $\beta$-VAE, pi-VAE and Swap-VAE compress neural activity independently for each time point. LFADS processes sequential neural activity with bidirectional RNNs. CEBRA encodes temporal features of sequence data with fixed convolutional kernels. None of them build latent representations progressively along the chronological order.

Furthermore, considering that our model has more parameters after incorporating the recurrent module, we build a small version of our model (TiDeSPL-VAE-small) with fewer trainable parameters than Swap-VAE (see Appendix E) for fair comparisons.

### 4.2 EXPERIMENTS ON SYNTHETIC DATA

We first validate TiDeSPL-VAE on the task of reconstructing synthetic latent variables. We generate two synthetic datasets for considering different properties of visual neural activity. One is a non-temporal dataset generated from several sets of labels, resembling categories of visual stimuli. The other is a temporal dataset constructed by the Lorenz system to test for time dependence. The generating procedure for both synthetic datasets follows some previous work (Zhou & Wei, 2020; Liu et al., 2021; Gao et al., 2016; Sussillo et al., 2016). A detailed description of the datasets and the model training implementation is presented in Appendix C. After training, we apply linear regression to map latent variables of models to the ground truth on the test set and report $R^2$ of the linear regression as the reconstruction score.

**Results on the non-temporal dataset** As shown in Figure 2, TiDeSPL-VAE reliably separates the different clusters as well as recovers the structure of true latent variables to form clear arcs. In contrast, some of the alternative models fail to construct similar structures although they separate clusters (Swap-VAE; CEBRA, Figure 6D of Appendix), and others even struggle to split four clusters ($\beta$-VAE, LFADS, and pi-VAE[1]; Figure 6A-C of Appendix). Quantitatively, the reconstruction scores also suggest that our model outperforms all alternative models (Figure 2D).

---

[1]pi-VAE incorporates the label prior during training, but inferred latent variables are built without the label prior at the evaluation stage. This way is used in all subsequent experiments.

Table 1: The reconstruction scores of all models on the synthetic temporal dataset. The standard error is computed based on 5 runs with different random initialization.

|  | $\beta$-VAE | LFADS | pi-VAE | Swap-VAE | CEBRA | **TiDeSPL-VAE** |
|---|---|---|---|---|---|---|
| Original | 0.036±0.027 | 0.573±0.043 | 0.167±0.012 | 0.193±0.004 | 0.242±0.001 | **0.629±0.016** |
| Shuffled | 0.025±0.022 | 0.020±0.006 | **0.209±0.007** | 0.146±0.008 | 0.051±0.005 | 0.038±0.007 |

**Results on the temporal dataset**  In the first row of Table 1, TiDeSPL-VAE performs significantly better than those models that process sequential data at each time point independently, and moderately better than LFADS that uses bidirectional RNNs to handle temporal data. These results demonstrate the superiority of our model in dealing with time-dependent data. Moreover, when we shuffle the time dimension for each trial data on the original dataset to obtain a dataset without time dependence (the second row of Table 1), the performance of our model and LFADS shows a drastic degradation. However, the other models are less affected, with only Swap-VAE and CEBRA suffering a degradation due to the use of time-jittered positive samples. This phenomenon further supports the above conclusions.

### 4.3 Experiments on Mouse Visual Cortex Data

We utilize a subset of the Allen Brain Observatory Visual Coding dataset (Siegle et al., 2021) for evaluation, which has been used in a variety of work, such as constructing brain-like networks (Shi et al., 2022), modeling functional mechanisms (Bakhtiari et al., 2021; de Vries et al., 2020), and decoding neural representations (Schneider et al., 2023). This dataset is collected by Neuropixel probes from 6 mouse visual cortical regions simultaneously, including VISp, VISl, VISrl, VISal, VISpm, and VISam. Notably, the neural activity was recorded while mice passively viewed naturalistic visual stimuli without any task-driven behavior.

The dataset contains 32 sessions, each for one mouse. Since the class of neurons responsive to natural visual stimuli is found in six visual regions, in this work we choose to analyze the neural activity of five mice that have the highest number of recorded neurons (see Appendix D for details), and these neurons are evenly distributed across all regions (the coefficient of variation for the number of neurons across six brain regions is below 0.5). We focus on neural activity in response to natural scenes and natural movies. As for natural scenes, there are 118 images presented in random order, each for 250ms and 50 trials. We select five scenes that elicit the strongest average responses for experiments. The neural activity in the form of spike counts is binned into 10ms windows so that each trial contains 25 time points. As for the natural movie, it is 30s long with a frame rate of 30Hz, presented for 10 trials. We bin the spike counts with a sampling frequency of 120Hz and align them with the movie timestamps, resulting in 4 time points for each frame. For both datasets, we randomly split each across all trials into 80% for training, 10% for validation, and 10% for test.

### 4.3.1 Experiments for Neural Activity under Static Natural Scene Stimuli

**Experiment setup and evaluation**  In this experiment, we set all models to have 128-dimensional latent variables and train them for 5,000 iterations. The optimizer is set to Adam with a learning rate of 0.0001. At the training stage, each sample input to TiDeSPL-VAE is sequential neural activity from 5 time points. For self-supervised contrastive learning, the offset of positive samples from target samples is within ±3 time points. At the inference stage, we consider the temporal dependence and the assumption that the latent variables of TiDeSPL-VAE are generated by an n-order Markov chain. Consequently, for a target time point, we form a sequence data including its antecedent n time points and its own to compute the latent variables (here n=4). The setup for the other models is given in Appendix F. To quantify performance in decoding natural scenes, we first obtain the latent variables of the last 20 time points (50ms-250ms) in each trial, since there is a response latency in the mouse visual cortex for static stimuli (Siegle et al., 2021). Then, these latent variables are concatenated into a vector as latent representations of neural activity for that trial. We use the KNN algorithm, a nonparametric supervised learning method, to classify the latent representations of each trial, i.e., to decode the corresponding natural scenes. We search the number of neighbors in odd numbers from 1 to 20 and use the Euclidean distance metric. We fit the KNN using the training set and choose the best number of neighbors on the validation set with classification accuracy as the metric. The accuracy of the test set is reported as the decoding score.

Table 2: The decoding scores (%) for natural scene classification on the visual neural dataset. The standard error is computed based on 10 runs with different random initialization.

| Models | Mouse 1 | Mouse 2 | Mouse 3 | Mouse 4 | Mouse 5 |
|---|---|---|---|---|---|
| PCA (baseline) | 20.0 | 43.2 | 51.2 | 43.6 | 20.0 |
| $\beta$-VAE | 57.6±4.3 | 47.6±1.4 | 33.6±1.8 | 46.4±2.3 | 44.4±2.2 |
| LFADS | 55.2±2.8 | 50.0±3.1 | 48.8±2.5 | 54.8±1.8 | 50.4±2.5 |
| pi-VAE | 76.4±3.4 | 67.2±2.5 | 68.8±2.3 | **81.2±2.5** | 30.4±2.2 |
| Swap-VAE | 86.0±1.6 | 70.8±2.0 | 54.4±1.5 | 67.2±2.9 | 59.6±3.5 |
| CEBRA | 47.6±2.3 | 46.0±1.6 | 42.8±2.0 | 52.4±1.7 | 45.2±1.5 |
| **TiDeSPL-VAE-small** | 90.0±2.0 | 65.2±3.2 | 71.2±2.2 | 74.4±2.7 | **70.8±2.0** |
| **TiDeSPL-VAE** | **96.4±1.1** | **74.8±2.0** | **74.8±1.7** | 78.8±2.9 | 67.6±2.1 |

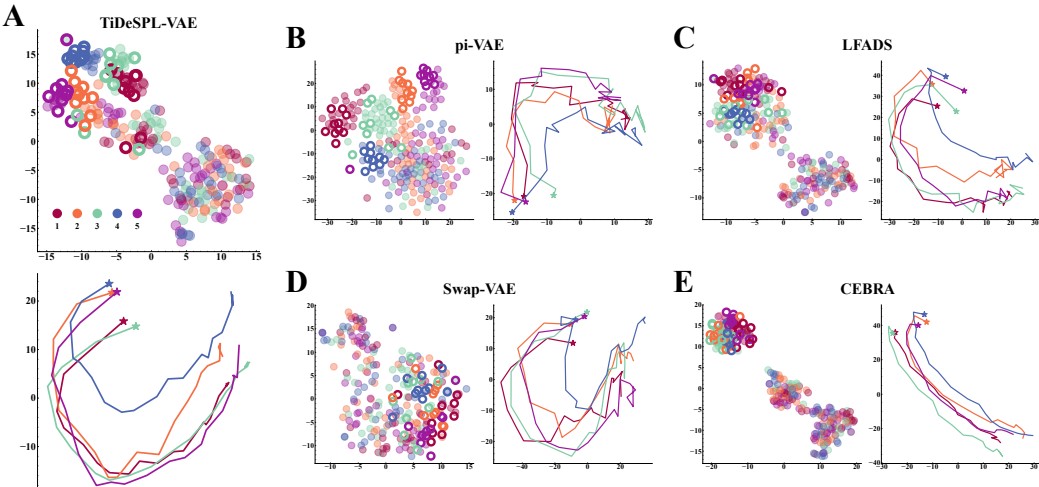

Figure 3: Results on the visual neural dataset under natural scenes (Mouse 1). All dimension reduction is done by tSNE. **A**. The upper panel is the 2-dimensional embedding of the latent representations from TiDeSPL-VAE for each trial. Each color corresponds to one natural scene. Transparent dots denote trials from the training set. Hollow dots denote trials from the test set. The lower panel is the 2-dimensional embedding of the latent trajectories over time, averaged over all trials for each scene. The asterisk is the starting point. **B-E**. The same visualization as **A** for alternative models.

**Results of decoding scores**    As shown in Table 2, TiDeSPL-VAE achieves the highest decoding scores on four mice with a noticeable improvement over other models. We observe some meaningful phenomena in the comparisons. The fact that pi-VAE introduces class labels of natural scenes in the training stage leads to high decoding scores, but not stably (Mouse 5). The performance of Swap-VAE is consistently good, which suggests that the swapping operation for training is indeed effective. However, the other two models (LFADS and CEBRA) that take sequential data as input instead perform poorly in the downstream task, suggesting that their approaches (bidirectional RNNs and temporal filters) for extracting temporal neural features are less suitable in this case. In contrast, the chronological stepwise computation in our model reliably captures time dependence.

**Results of latent structures**    In addition to the quantitative analysis, we visualize the latent representations by embedding them in two dimensions using tSNE (Figure 3, we focus on Mouse 1 with the highest scores. See Appendix H for results of other mice). On the one hand, we apply dimension reduction to the representations of each trial (including all time points from 50ms to 250ms), to show the trial-to-trial performance in decoding natural scenes. On the other hand, we reduce the dimensions of a single time point and take the average across trials, to show the latent trajectories over time. For the results of TiDeSPL-VAE, we first observe that most trials are well separated for different classes, especially those from the test set. The latent trajectories capture a similar clear temporal structure with weak class information. For pi-VAE and Swap-VAE, while they are able to discriminate scene classes of trials, the latter part of latent trajectories show varying degrees of

Table 3: The decoding scores (%, in 1s window) for natural movie frame classification on the visual neural dataset. The standard error is computed based on 10 runs with different random initialization.

| Models | Mouse 1 | Mouse 2 | Mouse 3 | Mouse 4 | Mouse 5 |
|---|---|---|---|---|---|
| PCA (baseline) | 8.44 | 28.77 | 25.42 | 21.56 | 11.69 |
| $\beta$-VAE | 7.44±0.24 | 15.13±0.35 | 14.00±0.37 | 17.11±0.50 | 9.28±0.31 |
| LFADS | 8.94±0.25 | 26.57±2.46 | 26.77±2.23 | 24.76±1.80 | 12.69±1.38 |
| pi-VAE | 10.24±0.31 | 42.51±0.65 | 36.96±0.60 | 38.31±0.52 | 18.08±0.59 |
| Swap-VAE | 11.09±0.25 | 44.99±0.76 | 36.37±1.53 | 37.68±1.14 | 19.14±0.63 |
| CEBRA | 10.62±0.18 | 52.76±0.89 | **61.01±0.76** | 42.11±0.73 | 22.33±0.31 |
| **TiDeSPL-VAE-small** | 12.26±0.30 | 63.30±0.34 | 57.57±0.39 | 53.46±0.58 | 28.70±0.42 |
| **TiDeSPL-VAE** | **13.88±0.19** | **65.38±0.36** | 59.88±0.72 | **54.33±0.54** | **30.18±0.40** |

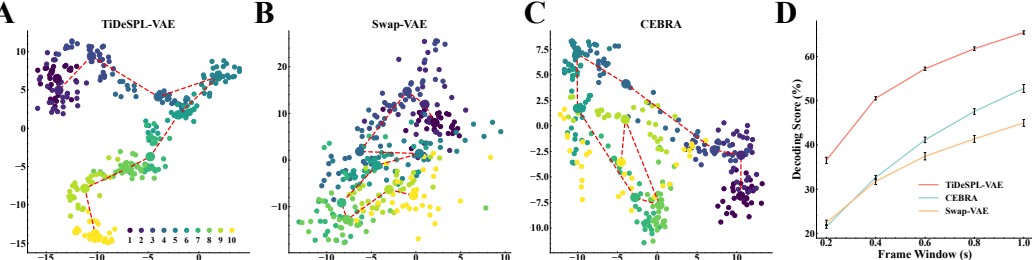

Figure 4: Results on the visual neural dataset under natural movie (Mouse 2). All dimension reduction is done by tSNE. **A**. The 2-dimensional embedding of the latent representations from TiDeSPL-VAE for each frame in the middle 10s of the entire movie. Each color corresponds to all frames within 1s. Small dots denote one frame. Large dots denote the average among all frames within the 1s. The red dashed line connects all averages. **B-C**. The same visualization as **A** for alternative models. **D**. The decoding scores (%) for natural movie frame classification across different time window constraints. Error bars indicate the standard error for 10 runs.

entanglement across time. As for LFADS and CEBRA, their ability to encode temporal features of sequential neural activity results in explicit temporal evolution, but the latent representations of different classes are largely intermingled. These results suggest that our model effectively distinguishes between category information and explicitly captures temporal information from neural dynamics.

In summary, these results demonstrate the advantages of our model in decoding natural scene stimuli and extracting fine neural dynamics. However, we also see that the decoding performance for different mice exhibits large differences and the latent representations of some trials are not able to unravel the category entanglement. This may be attributed to the substantial variability in the neural activity of different mice and different trials during passive viewing.

### 4.3.2 EXPERIMENTS FOR NEURAL ACTIVITY UNDER NATURAL MOVIE STIMULI

**Experiment setup and evaluation**  In this experiment, all models are also set up with 128-dimensional latent variables and trained for 20,000 iterations. The optimizer and the learning rate are the same as for the dataset under natural scenes. For TiDeSPL-VAE, at the training stage, we use neural activity from 4 time points as one sample and set the maximum absolute offset to 2. At the inference stage, following the approach in Section 4.3.1, the latent variables of a target time point are generated based on antecedent n time points (n=3) and its own. The setup for the other models is also presented in Appendix F. To quantify the performance in decoding natural movie frames, we compute the latent variables of 4 time points within each frame and take the average as latent representations for that frame. KNN is applied to predict movie frames corresponding to latent representations (900 frames in total, i.e., 900 classes). Following a previous method (Schneider et al., 2023), we take the accuracy measured by considering the error between a predicted frame and the true frame within 1s (default size of time window constraint) as a correct prediction. We similarly use the validation set to find the best number of neighbors and report the accuracy of the test set as the decoding score.

Table 4: The decoding scores (%) of ablation studies on the loss function and the recurrent module of TiDeSPL-VAE. The standard error is computed based on 10 runs.

| Models | Natural Scenes | | Natural Movie | |
|---|---|---|---|---|
| | Mouse 1 | Mouse 2 | Mouse 1 | Mouse 2 |
| TiDeSPL-VAE | **96.4±1.1** | **74.8±2.0** | **13.88±0.19** | **65.38±0.36** |
| Without negative samples | 94.0±1.7 | 70.4±0.8 | 11.27±0.36 | 49.59±1.18 |
| Without contrastive loss | 89.2±1.5 | 68.8±1.9 | 11.24±0.23 | 47.98±0.90 |
| Without swap operation | 90.4±1.7 | 68.0±2.7 | 10.22±0.31 | 49.39±0.57 |
| Without swap operation and constrastive loss | 84.4±2.6 | 58.0±2.1 | 9.16±0.37 | 24.84±1.10 |
| With temporal independent prior | 87.2±1.9 | 70.8±3.3 | 12.09±0.16 | 57.74±0.62 |
| GRU→Vanilla RNN | 90.4±1.5 | 69.6±3.1 | 13.08±0.31 | 63.19±0.55 |
| GRU→LSTM | 91.2±2.5 | 68.4±2.4 | 12.77±0.25 | 64.69±0.53 |
| Non-recurrent | 82.0±2.2 | 52.0±3.1 | 11.14±0.30 | 53.26±0.48 |

**Results of decoding scores**  We find that TiDeSPL-VAE performs best on four mice (Table 3). In particular, our model achieves significantly higher decoding scores than LFADS. Furthermore, we analyze the decoding scores under different sizes of time window constraints (maximum difference between predicted and real frames) on Mouse 2, since most models achieve the highest scores on it. Figure 4D shows that our model consistently outperforms CEBRA and Swap-VAE across a wide range of window sizes, while CEBRA's performance degrades faster at smaller size constraints.

**Results of latent structures**  We similarly reduce the dimensions of latent representations of each movie frame for visualization. We set all frames within 1s as a group and show the trajectories of latent representations evolving over time for the middle 10s of the movie (Figure 4A-C, see Appendix H for results from other parts of the movie). Compared to the other models, the representations of TiDeSPL-VAE show clear temporal structure along movie frames and less overlap and entanglement between different groups.

To conclude, our model constructs meaningful latent representations related to the content and temporal structure of movie stimuli at large time scales. At small time scales, although there is a drop in decoding performance, our model still significantly outperforms the alternatives. Similar features of adjacent frames may be a factor that makes the latter problem challenging, which may require further exploration.

## 4.4  ABLATION STUDIES

We perform ablation studies in several aspects of TiDeSPL-VAE's components and neural activity input dimensions to explore their impact on performance. We present the results of Mouse 1 and Mouse 2 in the main text and the results of the other mice are given in Appendix I.

**The loss function and the recurrent module of TiDeSPL-VAE**  To show the effectiveness of components of our model, we conduct some ablation studies on the loss function and the recurrent module (Table 4 and 9). In terms of contrastive learning, we first exclude negative samples from the computation of the contrastive loss and use only the cosine distance between the content latent variables of the positive sample pairs as the loss function, i.e., only bring the positive pairs closer. There is a slight decrease in model performance, suggesting that negative samples are useful. Then we directly remove the constrastive loss or the swap operation, both of which show a similar impact. We last remove both, i.e., there are no more objectives related to contrastive learning. The large drop in performance suggests that contrastive learning plays a crucial role in our models. In terms of the regular loss of style latent variables, we originally assumed that the prior distribution is time-dependent. When we assume that it is an independent standard normal distribution at each time step, the model performance degrades, demonstrating that time-dependent assumptions about the prior are also important. In terms of the recurrent module, the results suggest that GRU is a better choice in considering the trade-off between performance and computational efficiency. Besides, we evaluate a non-recurrent version of our model by setting the time steps of GRU to 1, demonstrating that the recurrent module plays a critical role.

Table 5: The decoding scores (%) of ablation studies on the content and style latent representations.

|  | Natural Scenes | | Natural Movie | |
|---|---|---|---|---|
|  | Mouse 1 | Mouse 2 | Mouse 1 | Mouse 2 |
| Content | **94.0±1.3** | **70.8±2.1** | **14.02±0.27** | **68.77±0.46** |
| Style | 76.4±1.9 | 62.0±2.5 | 7.48±0.20 | 14.87±0.73 |

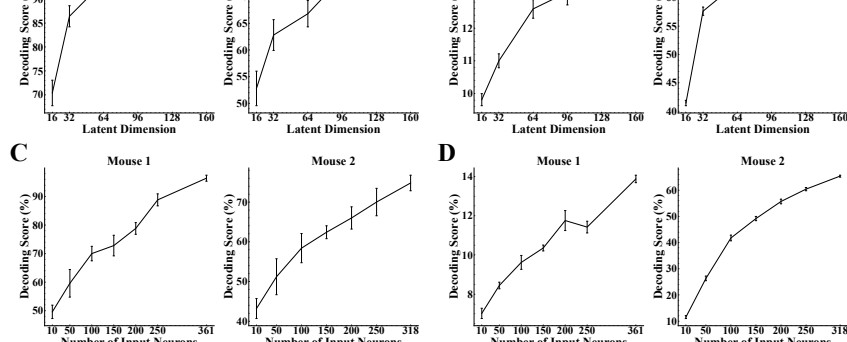

Figure 5: The results of ablation studies on the dimension of latent variables and the number of input neurons. Error bars indicate the standard error for 10 runs.

**The content and style latent representations**   We evaluate decoding scores of content and style latent representations (Table 5). We find that content variables outperform style variables, supporting our conceptual interpretation of them, i.e., content variables are more relevant to visual stimuli.

**The dimension of latent variables and the number of input neurons**   We perform ablation studies on the number of latent variables and input neurons. As shown in Figure 5 and 9, first, the model performance saturates gradually as the dimension of latent variables increases, which suggests that it is sufficient to choose a dimension at reasonable intervals (not too much fewer than input neurons). Second, performance decreases as the number of sampled neurons decreases, suggesting that for each mouse, all recorded neurons contribute to the representation of visual stimuli.

## 5 DISCUSSION

This work presents a novel sequential VAE by introducing time dependence into a self-supervised generative model, aiming to reveal intrinsic correlations between visual neural activity and visual stimulation. Our model, TiDeSPL-VAE, constructs latent representations conditioned on antecedent input to extract temporal relationships from neural activity in a natural way. Results of synthetic and mouse datasets demonstrate that our model outperforms alternative models and builds latent representations that are strongly correlated with visual stimulation, in terms of decoding natural scenes or movie frames and extracting explicit temporal structure from neural dynamics.

Most LVMs have focused on neural activity in the motor brain regions. There is a paucity of research explaining visual neural activity with this type of model. CEBRA has made meaningful explorations of neural responses of the mouse visual cortex (Schneider et al., 2023). Our work provides evidence that introducing time dependence with a chronological order plays a crucial role in studying visual neural activity with LVMs. However, there are some problems requiring further exploration. For example, neural responses elicited by complex natural visual stimuli show large variability across subjects and trials (Xia et al., 2021; Marks & Goard, 2021). Due to this, it is difficult for LVMs to consistently construct high-quality stimulus-correlated latent representations across conditions. Improvements in model structures and learning algorithms are needed in future work.

Last but not least, our approach is not limited to visual neural spikes from mice and can be extended to neural data from other brain regions and other species and of other modalities. As a promising model for neuroscience, it may also provide some insights into the field of machine learning.

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

## A  BACKBONE STRUCTURE OF TIDESPL-VAE

The encoder and decoder of our model are derived from Swap-VAE. The encoder consists of three blocks, the first two of which are sequentially stacked with a linear layer, a batch normalization, and a ReLU activation. The last block differs from Swap-VAE in that it additionally introduces the hidden states of the GRU as input. The output dimensions of the three blocks are $N$, $M$, and $M$, where $N$ is the number of neurons and $M$ is the number of latent variables. The decoder is a symmetric structure of the encoder, where the first two blocks are also sequentially stacked with a linear layer, a batch normalization, and a ReLU activation, and the last block is a linear layer followed by a SoftPlus activation. We set the dimensions of the content and style variables to be equal and use a one-layer GRU for each latent representation, where the dimensions of the hidden states are equal to the dimensions of the latent variables. For $\beta$-VAE, we use the same backbone structure. For other alternative models, we retain the structure of the original.

In the training, we ensure that the hyperparameters of all models are consistent, obtained by grid search. All models are trained on NVIDIA A100.

## B  DERIVATION OF THE LOSS FUNCTION OF TIDESPL-VAE

Given that we use state factors and recurrent neural networks to build time dependence in VAE and process sequential data, we need to maximize the likelihood of the joint sequential distribution $p(\mathbf{x}_{1:T})$. Involving the latent variables $\mathbf{z}_{1:T}$, we have the variational lower bound:

$$
\begin{aligned}
\log p(\mathbf{x}_{1:T}) &= \int q(\mathbf{z}_{1:T}|\mathbf{x}_{1:T}) \log p(\mathbf{x}_{1:T}) d\mathbf{z}_{1:T} \\
&= \int q(\mathbf{z}_{1:T}|\mathbf{x}_{1:T}) \log \frac{p(\mathbf{x}_{1:T}, \mathbf{z}_{1:T})}{p(\mathbf{z}_{1:T}|\mathbf{x}_{1:T})} d\mathbf{z}_{1:T} \\
&= \int q(\mathbf{z}_{1:T}|\mathbf{x}_{1:T}) \log \frac{q(\mathbf{z}_{1:T}|\mathbf{x}_{1:T})}{p(\mathbf{z}_{1:T}|\mathbf{x}_{1:T})} d\mathbf{z}_{1:T} + \int q(\mathbf{z}_{1:T}|\mathbf{x}_{1:T}) \log \frac{p(\mathbf{x}_{1:T}, \mathbf{z}_{1:T})}{q(\mathbf{z}_{1:T}|\mathbf{x}_{1:T})} d\mathbf{z}_{1:T} \\
&= \mathrm{KL}(q(\mathbf{z}_{1:T}|\mathbf{x}_{1:T})\|p(\mathbf{z}_{1:T}|\mathbf{x}_{1:T})) + \int q(\mathbf{z}_{1:T}|\mathbf{x}_{1:T}) \log \frac{p(\mathbf{x}_{1:T}, \mathbf{z}_{1:T})}{q(\mathbf{z}_{1:T}|\mathbf{x}_{1:T})} d\mathbf{z}_{1:T} \\
&\geq \int q(\mathbf{z}_{1:T}|\mathbf{x}_{1:T}) \log \frac{p(\mathbf{x}_{1:T}, \mathbf{z}_{1:T})}{q(\mathbf{z}_{1:T}|\mathbf{x}_{1:T})} d\mathbf{z}_{1:T},
\end{aligned}
\tag{8}
$$

where $p(\mathbf{x}_{1:T}, \mathbf{z}_{1:T})$ is the joint distribution as well as $p(\mathbf{z}_{1:T}|\mathbf{x}_{1:T})$ and $q(\mathbf{z}_{1:T}|\mathbf{x}_{1:T})$ is the true posterior and the variational approximate posterior, respectively. The true posterior is intractable.

Considering Eq. 1, Eq. 5 and Eq. 6, we know that $\mathbf{z}_t^{(c)}$ is deterministic values and $\mathbf{h}_t^{(s)}$ is a function of $\mathbf{x}_{1:t}$ and $\mathbf{z}_{1:t}^{(s)}$. Therefore, we have the factorization:

$$
p(\mathbf{x}_{1:T}, \mathbf{z}_{1:T}) = \prod_{t=1}^{T} p(\mathbf{x}_t|\mathbf{z}_{1:t}^{(s)}, \mathbf{z}_{1:t}^{(c)}, \mathbf{x}_{1:t-1}) p(\mathbf{z}_t^{(s)}|\mathbf{x}_{1:t-1}, \mathbf{z}_{1:t-1}^{(s)}),
\tag{9}
$$

$$
q(\mathbf{z}_{1:T}|\mathbf{x}_{1:T}) = \prod_{t=1}^{T} q(\mathbf{z}_t^{(s)}|\mathbf{x}_{1:t}, \mathbf{z}_{1:t-1}^{(s)}),
\tag{10}
$$

where $q(\mathbf{z}_t^{(s)}|\mathbf{x}_{1:t}, \mathbf{z}_{1:t-1}^{(s)})$, $p(\mathbf{z}_t^{(s)}|\mathbf{x}_{1:t-1}, \mathbf{z}_{1:t-1}^{(s)})$ and $p(\mathbf{x}_t|\mathbf{z}_{1:t}^{(s)}, \mathbf{z}_{1:t}^{(c)}, \mathbf{x}_{1:t-1})$ are the distributions defined by Eq. 2, Eq. 3 and Eq. 4, respectively. Based on the above factorization, we decompose the variational lower bound as:

$$\int q(\mathbf{z}_{1:T}|\mathbf{x}_{1:T}) \log \frac{p(\mathbf{x}_{1:T}, \mathbf{z}_{1:T})}{q(\mathbf{z}_{1:T}|\mathbf{x}_{1:T})} d\mathbf{z}_{1:T}$$

$$= \int q(\mathbf{z}_{1:T}|\mathbf{x}_{1:T}) \sum_{t=1}^{T} \left( \log \frac{p(\mathbf{x}_t|\mathbf{z}_{1:t}^{(s)}, \mathbf{z}_{1:t}^{(c)}, \mathbf{x}_{1:t-1}) p(\mathbf{z}_t^{(s)}|\mathbf{x}_{1:t-1}, \mathbf{z}_{1:t-1}^{(s)})}{q(\mathbf{z}_t^{(s)}|\mathbf{x}_{1:t}, \mathbf{z}_{1:t-1}^{(s)})} \right) d\mathbf{z}_{1:T} \qquad (11)$$

$$= \sum_{t=1}^{T} \left( \int q(\mathbf{z}_{1:T}|\mathbf{x}_{1:T}) \log \frac{p(\mathbf{x}_t|\mathbf{z}_{1:t}^{(s)}, \mathbf{z}_{1:t}^{(c)}, \mathbf{x}_{1:t-1}) p(\mathbf{z}_t^{(s)}|\mathbf{x}_{1:t-1}, \mathbf{z}_{1:t-1}^{(s)})}{q(\mathbf{z}_t^{(s)}|\mathbf{x}_{1:t}, \mathbf{z}_{1:t-1}^{(s)})} d\mathbf{z}_{1:T} \right).$$

When we simplify the above log-likelihood to a function $g(\mathbf{x}_{1:t}, \mathbf{z}_{1:t})$, we have:

$$\int q(\mathbf{z}_{1:T}|\mathbf{x}_{1:T}) g(\mathbf{x}_{1:t}, \mathbf{z}_{1:t}) d\mathbf{z}_{1:T}$$

$$= \int \left( \int q(\mathbf{z}_{1:T-1}|\mathbf{x}_{1:T-1}) q(\mathbf{z}_T^{(s)}|\mathbf{x}_{1:T}, \mathbf{z}_{1:T-1}^{(s)}) g(\mathbf{x}_{1:t}, \mathbf{z}_{1:t}) d\mathbf{z}_T \right) d\mathbf{z}_{1:T-1}$$

$$= \int \left( q(\mathbf{z}_{1:T-1}|\mathbf{x}_{1:T-1}) g(\mathbf{x}_{1:t}, \mathbf{z}_{1:t}) \int q(\mathbf{z}_T^{(s)}|\mathbf{x}_{1:T}, \mathbf{z}_{1:T-1}^{(s)}) d\mathbf{z}_T \right) d\mathbf{z}_{1:T-1} \qquad (12)$$

$$= \int q(\mathbf{z}_{1:T-1}|\mathbf{x}_{1:T-1}) g(\mathbf{x}_{1:t}, \mathbf{z}_{1:t}) d\mathbf{z}_{1:T-1}$$

$$= \cdots = \int q(\mathbf{z}_{1:t}|\mathbf{x}_{1:t}) g(\mathbf{x}_{1:t}, \mathbf{z}_{1:t}) d\mathbf{z}_{1:t}.$$

Therefore, we further decompose Eq. 11 as:

$$\int q(\mathbf{z}_{1:T}|\mathbf{x}_{1:T}) \log \frac{p(\mathbf{x}_{1:T}, \mathbf{z}_{1:T})}{q(\mathbf{z}_{1:T}|\mathbf{x}_{1:T})} d\mathbf{z}_{1:T}$$

$$= \sum_{t=1}^{T} \left( \int q(\mathbf{z}_{1:t}|\mathbf{x}_{1:t}) \log \frac{p(\mathbf{x}_t|\mathbf{z}_{1:t}^{(s)}, \mathbf{z}_{1:t}^{(c)}, \mathbf{x}_{1:t-1}) p(\mathbf{z}_t^{(s)}|\mathbf{x}_{1:t-1}, \mathbf{z}_{1:t-1}^{(s)})}{q(\mathbf{z}_t^{(s)}|\mathbf{x}_{1:t}, \mathbf{z}_{1:t-1}^{(s)})} d\mathbf{z}_{1:t} \right)$$

$$= \sum_{t=1}^{T} \left( \int q(\mathbf{z}_{1:t}|\mathbf{x}_{1:t}) \log p(\mathbf{x}_t|\mathbf{z}_{1:t}^{(s)}, \mathbf{z}_{1:t}^{(c)}, \mathbf{x}_{1:t-1}) d\mathbf{z}_{1:t} + \right.$$

$$\left. \int q(\mathbf{z}_{1:t}|\mathbf{x}_{1:t}) \log \frac{p(\mathbf{z}_t^{(s)}|\mathbf{x}_{1:t-1}, \mathbf{z}_{1:t-1}^{(s)})}{q(\mathbf{z}_t^{(s)}|\mathbf{x}_{1:t}, \mathbf{z}_{1:t-1}^{(s)})} d\mathbf{z}_{1:t} \right)$$

$$= \sum_{t=1}^{T} \left( \int q(\mathbf{z}_{1:t}|\mathbf{x}_{1:t}) \log p(\mathbf{x}_t|\mathbf{z}_{1:t}^{(s)}, \mathbf{z}_{1:t}^{(c)}, \mathbf{x}_{1:t-1}) d\mathbf{z}_{1:t} - \right. \qquad (13)$$

$$\left. \int q(\mathbf{z}_{1:t-1}|\mathbf{x}_{1:t-1}) \mathrm{KL}(q(\mathbf{z}_t^{(s)}|\mathbf{x}_{1:t}, \mathbf{z}_{1:t-1}^{(s)}) \| p(\mathbf{z}_t^{(s)}|\mathbf{x}_{1:t-1}, \mathbf{z}_{1:t-1}^{(s)})) d\mathbf{z}_{1:t-1} \right)$$

$$= \int q(\mathbf{z}_{1:T}|\mathbf{x}_{1:T}) \sum_{t=1}^{T} \left( \log p(\mathbf{x}_t|\mathbf{z}_{1:t}^{(s)}, \mathbf{z}_{1:t}^{(c)}, \mathbf{x}_{1:t-1}) - \right.$$

$$\left. \mathrm{KL}(q(\mathbf{z}_t^{(s)}|\mathbf{x}_{1:t}, \mathbf{z}_{1:t-1}^{(s)}) \| p(\mathbf{z}_t^{(s)}|\mathbf{x}_{1:t-1}, \mathbf{z}_{1:t-1}^{(s)})) \right) d\mathbf{z}_{1:T}$$

$$= \mathbb{E}_{q(\mathbf{z}_{1:T}|\mathbf{x}_{1:T})} \left[ \sum_{t=1}^{T} \log p(\mathbf{x}_t|\mathbf{z}_{1:t}^{(s)}, \mathbf{z}_{1:t}^{(c)}, \mathbf{x}_{1:t-1}) - \right.$$

$$\left. \mathrm{KL}(q(\mathbf{z}_t^{(s)}|\mathbf{x}_{1:t}, \mathbf{z}_{1:t-1}^{(s)}) \| p(\mathbf{z}_t^{(s)}|\mathbf{x}_{1:t-1}, \mathbf{z}_{1:t-1}^{(s)})) \right].$$

Finally, for a given sequential data x, we have the loss function for training the generative objective:

$$\mathcal{L} \simeq \sum_{t=1}^{T} \left( \underbrace{- \log p(\mathbf{x}_t | \mathbf{z}_{1:t}^{(s)}, \mathbf{z}_{1:t}^{(c)}, \mathbf{x}_{1:t-1})}_{\text{reconstruction loss}} + \underbrace{\mathrm{KL}(q(\mathbf{z}_t^{(s)} | \mathbf{x}_{1:t}, \mathbf{z}_{1:t-1}^{(s)}) \| p(\mathbf{z}_t^{(s)} | \mathbf{x}_{1:t-1}, \mathbf{z}_{1:t-1}^{(s)}))}_{\text{regularization loss}} \right),$$

(14)

where the first and second terms correspond to $\mathcal{L}_{\mathrm{P}}$ and $\mathrm{D}_{\mathrm{KL}}$ in the main text, respectively.

Since we assume a Poisson distribution for the reconstructed neural activity, $\mathcal{L}_{\mathrm{P}}$ is the Poisson negative log-likelihood:

$$\begin{aligned}
\mathcal{L}_{\mathrm{P}}(\mathbf{x}_t, \mathbf{r}_t) &= - \log \frac{\mathbf{r}_t^{\mathbf{x}_t}}{\mathbf{x}_t!} e^{-\mathbf{r}_t} \\
&= -\mathbf{x}_t \log \mathbf{r}_t + \mathbf{r}_t + \log \mathbf{x}_t! \\
&\approx -\mathbf{x}_t \log \mathbf{r}_t + \mathbf{r}_t + \mathbf{x}_t \log \mathbf{x}_t - \mathbf{x}_t + \frac{1}{2} \log (2\pi \mathbf{x}_t).
\end{aligned}$$

(15)

As for $\mathrm{D}_{\mathrm{KL}}$, under the assumption that both the prior and the approximate posterior are Gaussian, we have:

$$\begin{aligned}
\mathrm{D}_{\mathrm{KL}}(\mathbf{z}_t^{(s)} \| \tilde{\mathbf{z}}_t^{(s)}) &= \int q(\mathbf{z}_t^{(s)} | \mathbf{x}_{1:t}, \mathbf{z}_{1:t-1}^{(s)}) \log \frac{q(\mathbf{z}_t^{(s)} | \mathbf{x}_{1:t}, \mathbf{z}_{1:t-1}^{(s)})}{p(\mathbf{z}_t^{(s)} | \mathbf{x}_{1:t-1}, \mathbf{z}_{1:t-1}^{(s)})} d\mathbf{z}_t^{(s)} \\
&= \int q(\mathbf{z}_t^{(s)} | \mathbf{x}_{1:t}, \mathbf{z}_{1:t-1}^{(s)}) \log \frac{\frac{1}{\sqrt{2\pi\boldsymbol{\sigma}_{z,t}^2}} \exp\left( -\frac{\left(\mathbf{z}_t^{(s)} - \boldsymbol{\mu}_{z,t}\right)^2}{2\boldsymbol{\sigma}_{z,t}^2} \right)}{\frac{1}{\sqrt{2\pi\tilde{\boldsymbol{\sigma}}_{z,t}^2}} \exp\left( -\frac{\left(\mathbf{z}_t^{(s)} - \tilde{\boldsymbol{\mu}}_{z,t}\right)^2}{2\tilde{\boldsymbol{\sigma}}_{z,t}^2} \right)} d\mathbf{z}_t^{(s)} \\
&= - \frac{\mathbb{E}_{q(\mathbf{z}_t^{(s)})}\left[ \left(\mathbf{z}_t^{(s)} - \boldsymbol{\mu}_{z,t}\right)^2 \right]}{2\boldsymbol{\sigma}_{z,t}^2} + \frac{\mathbb{E}_{q(\mathbf{z}_t^{(s)})}\left[ \left(\mathbf{z}_t^{(s)} - \tilde{\boldsymbol{\mu}}_{z,t}\right)^2 \right]}{2\tilde{\boldsymbol{\sigma}}_{z,t}^2} - \log \boldsymbol{\sigma}_{z,t} + \log \tilde{\boldsymbol{\sigma}}_{z,t} \\
&= - \frac{1}{2} + \frac{\boldsymbol{\sigma}_{z,t}^2 + \boldsymbol{\mu}_{z,t}^2 - 2\boldsymbol{\mu}_{z,t}\tilde{\boldsymbol{\mu}}_{z,t} + \tilde{\boldsymbol{\mu}}_{z,t}^2}{2\tilde{\boldsymbol{\sigma}}_{z,t}^2} - \log \boldsymbol{\sigma}_{z,t} + \log \tilde{\boldsymbol{\sigma}}_{z,t} \\
&= \frac{1}{2} \left( -1 + \frac{\left(\boldsymbol{\mu}_{z,t} - \tilde{\boldsymbol{\mu}}_{z,t}\right)^2 + \boldsymbol{\sigma}_{z,t}^2}{\tilde{\boldsymbol{\sigma}}_{z,t}^2} - \log \boldsymbol{\sigma}_{z,t}^2 + \log \tilde{\boldsymbol{\sigma}}_{z,t}^2 \right).
\end{aligned}$$

(16)

Finally, we apply NT-Xent loss as the contrastive loss:

$$\mathcal{L}_{\text{contrast}} = - \log \frac{\exp\left( \mathrm{sim}\left( \mathbf{z}^{(c)}, \mathbf{z}_{\text{pos}}^{(c)} \right) / \tau \right)}{\exp\left( \mathrm{sim}\left( \mathbf{z}^{(c)}, \mathbf{z}_{\text{pos}}^{(c)} \right) / \tau \right) + \sum \exp\left( \mathrm{sim}\left( \mathbf{z}^{(c)}, \mathbf{z}_{\text{neg}}^{(c)} \right) / \tau \right)},$$

(17)

where $\mathrm{sim}(*, *)$ is the cosine similarity and $\tau$ is the temperature coefficient.

## C  SYNTHETIC DATASETS

### C.1  GENERATING PROCEDURE

**Non-temporal dataset**  First, we generate labels $u_i$ from four uniform distributions on $[\frac{2i \times \pi}{4}, \frac{(2i+1) \times \pi}{4}], i \in \{0, 1, 2, 3\}$, in preparation for building four clusters. Second, for each cluster, we sample 2-dimensional latent variables $\mathbf{z}$ from independent Gaussian distribution with

$(5 \sin u_i, 5 \cos u_i)$ as mean and $(0.6 - 0.5 |\cos u_i|, 0.5 |\cos u_i|)$ as variance. Third, we feed sampled latent variables into a RealNVP network (Dinh et al., 2017) to form firing rates of 100-dimensional observations and generate the synthetic neural activity from the Poisson distribution. Each cluster contains 4,000 samples. All samples are split to 80% (12,800 samples) for training and 20% (3,200 samples) for test.

**Temporal dataset** We generate three dynamic latent variables from the Lorenz system consisting of a set of nonlinear equations and the firing rates of 30 simulated neurons are computed by randomly weighted linear readouts from the Lorenz latent variables. The synthetic neural activity is also generated from the Poisson distribution. The hyperparameters of the Lorenz system follow some previous work (Sussillo et al., 2016). We run the Lorenz system for 1s (1ms for a time point) from five randomly initialized conditions. Each condition contains 20 trials. We use 80% of the dataset (16 trials/condition, 80,000 samples) for training and 20% (4 trials/condition, 20,000 samples) for test.

## C.2 TRAINING IMPLEMENTATION

**Non-temporal dataset** All models are set up with 32-dimensional latent variables and trained for 20,000 iterations with an optimizer of Adam and a learning rate of 0.0005. Notably, since this dataset does not involve time dependence, the length of input sequences is set to 1 even for models that can handle sequential data.

**Temporal dataset** All models are set up with 8-dimensional latent variables and trained for 20,000 iterations with an optimizer of Adam and a learning rate of 0.001. Our model and LFADS use 50ms of data as input, while the other models take data at one time point because they can't process sequential data.

## D CHARACTERS OF NEURAL DATASET

In this work, we use a subset of the Allen Brain Observatory Visual Coding dataset (de Vries et al., 2020; Siegle et al., 2021) recorded from six visual cortical regions of the mouse with Neuropixel probes. The full names and abbreviations of all cortical regions are listed in Table 6. We present the number of neurons for all chosen mice.

Table 6: Characters of the neural dataset.

| Cortical Region | Abbreviation | Mouse 1 | Mouse 2 | Mouse 3 | Mouse 4 | Mouse 5 |
|---|---|---|---|---|---|---|
| primary visual cortex | VISp | 75 | 51 | 93 | 63 | 52 |
| lateromedial area | VISl | 39 | 30 | 56 | 38 | 20 |
| rostrolateral area | VISrl | 49 | 24 | 58 | 44 | 41 |
| anterolateral area | VISal | 42 | 51 | 43 | 71 | 46 |
| posteromedial area | VISpm | 62 | 90 | 17 | 19 | 64 |
| anteromedial area | VISam | 94 | 72 | 49 | 60 | 64 |

In our experiments, we focus on neural activity in response to natural scenes and natural movies. For both datasets, we randomly split each across all trials into 80% for training, 10% for validation, and 10% for test. The number of samples for each dataset is shown in Table 7.

Table 7: The number of samples for the visual neural dataset.

| Dataset | Training | Validation | Test |
|---|---|---|---|
| visual neural dataset under natural scenes | 5000 | 625 | 625 |
| visual neural dataset under natural movies | 28800 | 3600 | 3600 |

## E NUMBER OF TRAINABLE PARAMETERS OF ALL MODELS

The number of model parameters is roughly proportional to the number of input neurons. We present the number of parameters for the Mouse 1 dataset.

Table 8: The number of model parameters for the Mouse 1 dataset.

| | $\beta$-VAE | LFADS | pi-VAE | Swap-VAE | CEBRA | TiDeSPL-VAE-small | TiDeSPL-VAE |
|---|---|---|---|---|---|---|---|
| Number of parameters | 0.39M | 0.45M | 0.49M | 0.38M | 0.71M | 0.29M | 0.68M |

## F EXPERIMENT SETUP ON NEURAL DATASET FOR ALTERNATIVE MODEL

For $\beta$-VAE, pi-VAE, and Swap-VAE, each input sample is neural activity of an independent time point, both at the training and inference stages. For LFADS, we apply the same approach as our model. For CEBRA, following the original approach (Schneider et al., 2023), we take the surrounding points centered on the target point to form a sequence of the same length as during the training stage (5 time points for natural scenes and 4 for natural movie) and compute its latent variables.

## G ADDITIONAL RESULTS ON SYNTHETIC NON-TEMPORAL DATASET

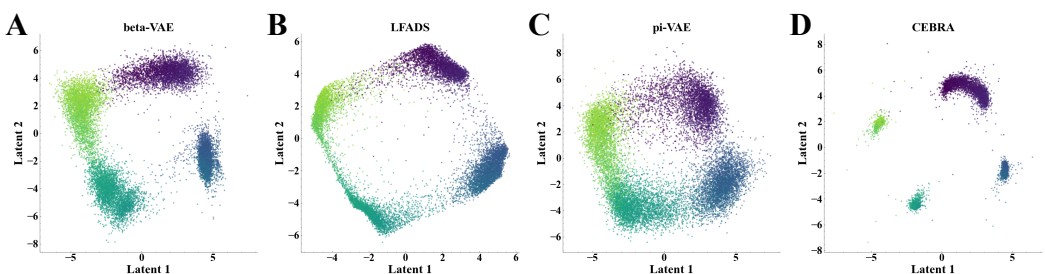

Figure 6: The regressed latent variables of alternatives on the synthetic non-temporal dataset.

## H ADDITIONAL VISUALIZATION OF LATENT REPRESENTATIONS

We visualize latent representations of TiDeSPL-VAE in experiments under natural scenes for the other four mice and latent representations of TiDeSPL-VAE, SwapVAE and CEBRA in experiments under natural movie stimuli for other parts of the movie. The results are in line with the conclusion in the main text.

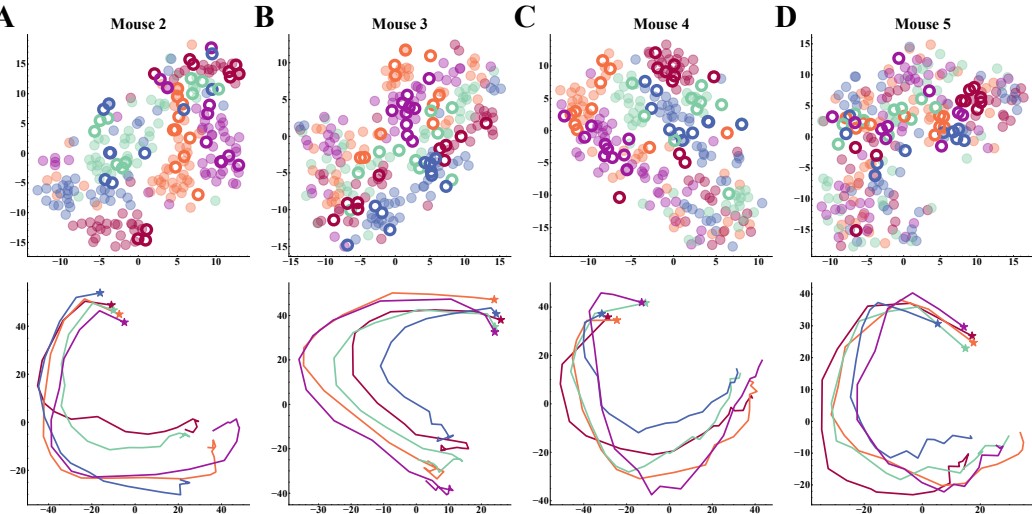

Figure 7: Visualization results of TiDeSPL-VAE on the visual neural dataset under natural scenes (Mouse 2-5).

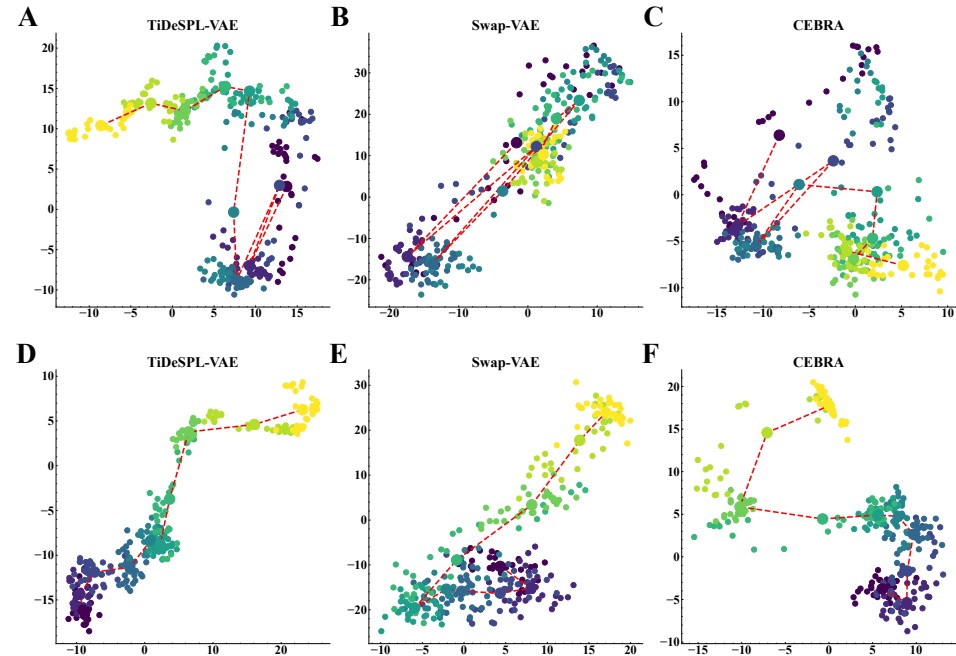

Figure 8: Visualization results on the visual neural dataset under natural movie (Mouse 2). **A-C**. The 2-dimensional embedding of the latent representations for each frame in the first 10s of the movie. **D-F**. The 2-dimensional embedding of the latent representations for each frame in the last 10s of the movie. The first 10s show more entanglement across time, while the last 10s are rarely entangled. Nevertheless, our model still outperforms other alternative models.

# I    ADDITIONAL RESULTS OF ABLATION STUDIES

We present the results of ablation studies for the other three mice, which are consistent with the conclusion in the main text.

Table 9: The decoding scores (%) of ablation studies on the loss function and the recurrent module of TiDeSPL-VAE. The standard error is computed based on 10 runs.

| Models | Natural Scenes | | | Natural Movie | | |
|---|---|---|---|---|---|---|
| | Mouse 3 | Mouse 4 | Mouse 5 | Mouse 3 | Mouse 4 | Mouse 5 |
| TiDeSPL-VAE | **74.8±1.7** | 78.8±2.9 | **67.6±2.1** | 59.88±0.72 | 54.33±0.54 | **30.18±0.40** |
| Without negative samples | 67.2±2.5 | 76.0±2.0 | 59.6±2.6 | 45.67±0.60 | 44.17±0.42 | 19.93±0.35 |
| Without contrastive loss | 72.0±1.6 | 76.4±2.6 | 60.4±1.9 | 44.09±0.67 | 44.02±0.47 | 18.24±0.41 |
| Without swap operation | 66.4±1.9 | 72.4±2.4 | 65.2±3.0 | 45.30±0.41 | 43.12±0.57 | 21.83±0.39 |
| Without swap operation and constrastive loss | 59.6±2.7 | 66.8±2.3 | 53.6±2.8 | 22.33±0.81 | 26.49±0.66 | 12.02±0.49 |
| With non-temporal prior | 64.4±2.6 | 69.6±3.0 | 61.2±3.0 | 53.87±0.49 | 46.90±0.48 | 22.92±0.43 |
| GRU→Vanilla RNN | 72.4±2.1 | 73.6±2.3 | 58.4±1.6 | 59.13±0.42 | 53.83±0.41 | 29.62±0.40 |
| GRU→LSTM | 70.8±1.5 | **80.8±1.9** | 62.4±2.4 | **60.00±0.60** | **54.37±0.41** | 28.50±0.61 |
| Non-recurrent | 56.4±2.7 | 68.0±2.6 | 59.6±4.1 | 48.31±0.47 | 43.81±0.40 | 22.98±0.44 |

Table 10: The decoding scores (%) of ablation studies on the content and style latent representations of TiDeSPL-VAE. The standard error is computed based on 10 runs.

| | Natural Scenes | | | Natural Movie | | |
|---|---|---|---|---|---|---|
| | Mouse 3 | Mouse 4 | Mouse 5 | Mouse 3 | Mouse 4 | Mouse 5 |
| Content | **76.4±1.6** | **77.6±2.3** | **68.0±2.3** | **65.92±0.51** | **57.07±0.44** | **32.39±0.48** |
| Style | 60.8±2.8 | 61.6±3.2 | 49.2±3.3 | 14.62±1.06 | 17.96±1.00 | 11.12±0.39 |

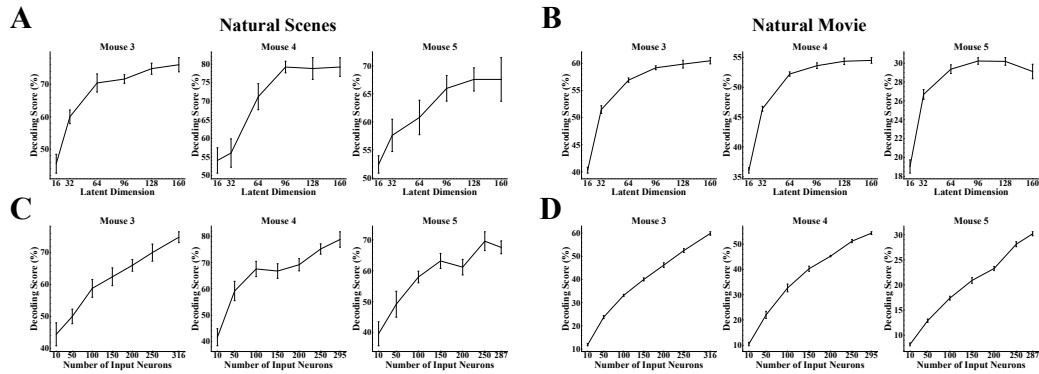

Figure 9: The results of ablation studies on the dimension of latent variables and the number of input neurons. Error bars indicate the standard error for 10 runs.

In addition, more ablation studies in other aspects of TiDeSPL-VAE are shown in Table 11 and 12. By removing the reconstruction loss and the KL divergence computed on positive samples, we find that these loss terms contribute less to performance, suggesting that the contrastive loss and the swap operation are sufficient to emphasize the positive samples. We evaluate two versions of our model containing only content or style latent representations. The results demonstrate that splitting these two latent representations is important and support the conclusion of Table 5 and 10.

Table 11: The decoding scores (%) of further ablation studies on the loss function and the latent representation of TiDeSPL-VAE for natural scene classification.

| Models | Mouse 1 | Mouse 2 | Mouse 3 | Mouse 4 | Mouse 5 |
|---|---|---|---|---|---|
| TiDeSPL-VAE | **96.4±1.1** | **74.8±2.0** | **74.8±1.7** | **78.8±2.9** | **67.6±2.1** |
| Without reconstruction and KL losses on positive samples | 96.0±1.7 | 71.2±1.8 | 74.4±1.4 | 77.2±1.4 | 65.6±1.9 |
| Containing only content | 74.0±2.0 | 49.6±2.8 | 53.2±2.3 | 68.8±3.8 | 56.0±3.1 |
| Containing only style | 60.0±3.0 | 32.4±2.6 | 44.4±2.6 | 47.2±1.7 | 44.8±2.8 |

Table 12: The decoding scores (%) of further ablation studies on the loss function and the latent representation of TiDeSPL-VAE for natural movie frame classification.

| Models | Mouse 1 | Mouse 2 | Mouse 3 | Mouse 4 | Mouse 5 |
|---|---|---|---|---|---|
| TiDeSPL-VAE | **13.88±0.19** | **65.38±0.36** | **59.88±0.72** | **54.33±0.54** | 30.18±0.40 |
| Without reconstruction and KL losses on positive samples | 12.80±0.19 | 64.18±0.42 | 58.70±0.53 | 54.00±0.39 | 29.97±0.50 |
| Containing only content | 12.16±0.24 | 63.33±0.29 | 58.64±0.37 | 52.11±0.39 | **30.24±0.54** |
| Containing only style | 7.57±0.24 | 11.73±0.43 | 12.86±0.35 | 16.10±0.33 | 9.89±0.22 |

## J ABLATION STUDIES ON THE MOUSE VISUAL NEURAL DATASET FOR TRAINING AND TEST

In the Allen Brain Observatory Visual Coding dataset, each mouse is implanted with Neuropixel probes separately, so there is no guarantee that the neural sites recorded from each mouse are aligned, and the number of recorded neurons is different. Therefore, we treat each mouse as a single dataset in the main experiments. For further analysis, we sample neurons from all five mice evenly to form a dataset (named All Mice) with a similar number of neurons to a single mouse dataset. The results in Table 13 show that our model outperforms other models on All Mice, but the performance is overall lower than that on a single mouse dataset, suggesting variability in neural activity to the same visual stimuli across mice.

Table 13: The decoding scores (%) on the All Mice datasets.

|  | LFADS | pi-VAE | Swap-VAE | CEBRA | **TiDeSPL-VAE** |
|---|---|---|---|---|---|
| All Mice (natural scene) | 41.6±4.0 | 44.0±3.8 | 34.4±4.9 | 46.4±3.7 | **49.6±2.4** |
| All Mice (natural movie) | 16.36±1.02 | 22.44±0.53 | 22.07±0.28 | 22.98±0.76 | **28.07±1.01** |

In addition, we sample the same number of neurons (250) in each mouse. We then train our model on one mouse and test it on the other. As shown in Table 14 and 15, for the same mouse in the test, there is a significant drop in performance for models trained on other mice. This may be due to the difficulty of aligning the recorded neurons and the large variation in response patterns between mice.

Table 14: The decoding scores (%) of TiDeSPL-VAE which is trained on one mouse and tested on the other for natural scene classification.

| Training \ Test | Mouse 1 | Mouse 2 | Mouse 3 | Mouse 4 | Mouse 5 |
|---|---|---|---|---|---|
| Mouse 1 | **88.8±2.1** | 48.0±2.2 | 59.2±3.2 | 60.0±3.0 | 50.4±2.6 |
| Mouse 2 | 54.0±3.9 | **70.0±3.5** | 51.6±3.7 | 64.0±3.1 | 48.0±2.9 |
| Mouse 3 | 56.0±2.9 | 48.8±3.4 | **70.0±2.7** | 58.0±3.4 | 50.4±2.1 |
| Mouse 4 | 63.2±4.0 | 53.2±3.3 | 51.6±1.7 | **75.2±1.9** | 56.0±2.9 |
| Mouse 5 | 64.4±2.3 | 42.8±1.6 | 42.4±2.7 | 54.4±3.1 | **69.6±3.1** |

Table 15: The decoding scores (%) of TiDeSPL-VAE which is trained on one mouse and tested on the other for natural movie frame classification.

| Training \ Test | Mouse 1 | Mouse 2 | Mouse 3 | Mouse 4 | Mouse 5 |
|---|---|---|---|---|---|
| Mouse 1 | **11.42±0.30** | 25.87±0.36 | 20.76±0.62 | 23.56±0.45 | 12.83±0.58 |
| Mouse 2 | 8.60±0.23 | **60.41±0.62** | 17.47±0.50 | 18.91±0.54 | 10.79±0.29 |
| Mouse 3 | 8.32±0.13 | 19.82±0.88 | **52.50±0.84** | 19.53±0.41 | 11.43±0.32 |
| Mouse 4 | 8.42±0.35 | 20.24±0.63 | 18.08±0.51 | **51.18±0.64** | 11.26±0.31 |
| Mouse 5 | 8.34±0.29 | 21.00±0.51 | 18.60±0.35 | 19.51±0.54 | **28.10±0.50** |

## K ABLATION STUDY ON NATURAL SCENE STIMULI

In addition to the experiments on the five scenes that elicit the strongest average responses, we select five other scenes that elicit the weakest responses for an ablation study. As shown in Table 16, compared to the experiments on the scenes that elicit the strongest responses, the performance on the scenes that elicit the weakest responses is lower overall, suggesting that it is more difficult for models to build stimulus-relevant latent variables from neural activity with low signal-to-noise. Nevertheless, our model still performs best in most cases, demonstrating its robustness.

Table 16: The decoding scores (%) for natural scene classification on the visual neural dataset under five scenes that elicit the weakest responses.

| Models | Mouse 1 | Mouse 2 | Mouse 3 | Mouse 4 | Mouse 5 |
|---|---|---|---|---|---|
| LFADS | 44.8±3.5 | 23.2±3.5 | 26.4±1.4 | 36.8±3.3 | 35.2±3.6 |
| pi-VAE | 20.0±2.5 | 28.8±3.1 | 29.6±2.7 | 20.8±0.7 | 19.2±0.7 |
| Swap-VAE | 47.2±2.6 | 32.8±3.1 | 26.4±0.9 | 33.6±3.3 | 28.8±1.3 |
| CEBRA | 36.0±3.0 | 27.2±2.1 | 29.6±1.4 | 32.8±1.3 | **36.0±1.1** |
| **TiDeSPL-VAE** | **58.4±2.7** | **35.2±3.5** | **36.8±2.4** | **42.4±2.1** | 32.0±1.6 |

## L  VISUALIZATION WITH DIFFERENT HYPERPARAMETERS OF tSNE

To analyze the visualization of our model's latent representations on the Mouse 1 dataset under natural scene stimuli, we select different hyperparameters of tSNE (*perplexity* in the range [5, 50] and *early_exaggeration* in the range [12, 24]). The results (Figure 10 and 11) show that the embedding properties are stable across different hyperparameters.

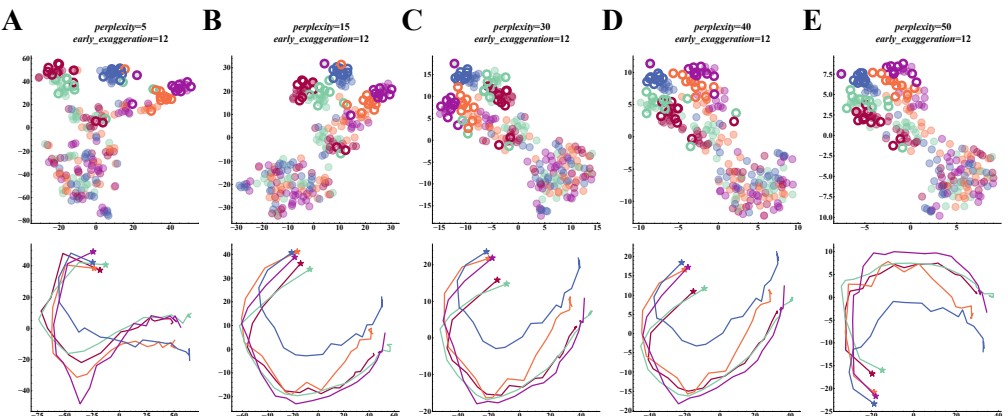

Figure 10: Visualization results of TiDeSPL-VAE on the Mouse 1 dataset under natural scene stimuli (different *perplexity* and same *early_exaggeration*).

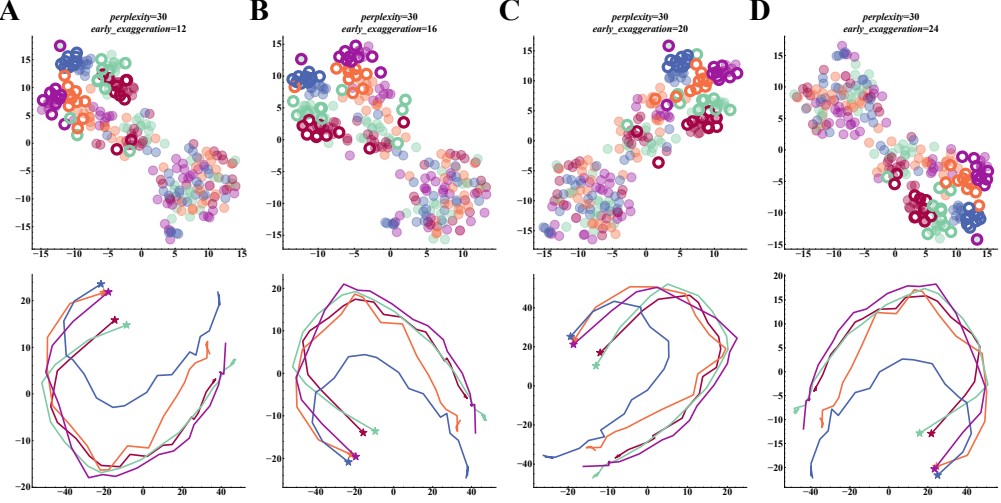

Figure 11: Visualization results of TiDeSPL-VAE on the Mouse 1 dataset under natural scene stimuli (same *perplexity* and different *early_exaggeration*).

