# OpenReview forum: "Time-Dependent VAE for Building Latent Representations from Visual Neural Activity with Complex Dynamics"
_ICLR.cc/2025/Conference — Submitted to ICLR 2025_

### Official Review · Reviewer_MX7e · 2024-10-16

**Soundness:** 3
**Presentation:** 3
**Contribution:** 3
**Rating:** 8
**Confidence:** 3

**Summary:**

The authors introduce the latent model, which separates the visual activity in the content and context representations, also taking the time dependency into the account. This seems to help to have more disentangles representations of mice visual cortex activities

**Strengths:**

* Original work, applying a new idea of splitting context and content of neural activity in the visual cortex
* The necessary baselines such as LFADS, pi-VAE, and CEBRA are represented in the paper
* Careful evaluation on both synthetic and real-world datasets, also following classic for the field benchmarks, like Lorenz systems
* Sanity checks experiments, such as shuffling time dependent dateset help to disentangle the impact of time-dependency for the model
* The authors are very transparent about the limitations of the model

**Weaknesses:**

* Reproducubility might be an issue. The code is not provided and the implementation details in the appendix A are not sufficient. Details like how many RNN layers were used, how exactly the RNN was implemented, etc are missing. GRU is mentioned later in the paper but I think an explicit model architecture picture and more parameter details in appendix A would help to improve the paper.
* Generalisability and robustness of the model is an issue as scores per mouse differ a lot, also it is not clear how the model would behave if the animal was shown different stimuli (ie gratings instead of the natural images).
* Some references like [1,2] are missing, which could be nice to position the TiDeSPL-VAE model in the space of LVM as they are the representatives of modern non-VAE based LVM (while they are different from the current work)

[1] Bashiri et al 2021 https://proceedings.neurips.cc/paper/2021/hash/84a529a92de322be42dd3365afd54f91-Abstract.html
[2] Kapoor, Schulz et al 2024 https://arxiv.org/pdf/2407.08751

**Questions:**

* RNNs are often tricky to train. Were there any issues that you phased during the model training or some details, which are crucial for a successful learning.
* Did you train a new separate model per mouse? Or was it a shared model across mice?
* In figure 3 for TiDeSPL-VAE, LFADS and especially CEBRA there seems to be a clear train/test bias (points are separated), which is stronger than the cluster separartion based on the stimuli. How do you interpret this?
* how exactly the train/test splits are done with respect to the time? As the train loss takes a sequence of responses from the near future with some $\vartriangle t$ - do you ensure that none of the validation/test responses were shown during training?

---

> ### Author Response · Authors · 2024-11-22
>
> We thank the reviewer for the perceptive and detailed comments. We will do our best to address the comments and answer the questions.
>
> **1. Reproducubility might be an issue.**
>
> **We have provided our code in the supplementary material, including the model, training and testing**. We use a one-layer GRU implemented by pytorch for each latent space, and the dimensions of the hidden states of the GRU are the same as the dimensions of the latent variables. We will provide an explicit figure of the model architecture and more detailed descriptions in the revision.
>
> **2. Generalisability and robustness of the model is an issue.**
>
> This problem may be due to the fact that neural responses elicited by complex natural visual stimuli show large variability across subjects [1] and we have discussed this limitation in lines 533-537. However, as suggested by the reviewer umrd, we choose five other scenes that elicit the weakest responses, while the experiments in our work are conducted on five scenes that elicit the strongest responses. We conduct two experiments, one with these new scenes only and one with all ten scenes. As shown in Tables R1 and R2, **our model performs best in most cases, demonstrating its robustness to some extent**. Besides, we will evaluate our model on different visual stimuli (e.g. gratings) in the future.
>
> |Model|Mouse 1|Mouse 2|Mouse 3|Mouse 4|Mouse 5|
> |-|:-:|:-:|:-:|:-:|:-:|
> |LFADS|44.8±3.5|23.2±3.5|26.4±1.4|36.8±3.3|35.2±3.6|
> |pi-VAE|20.0±2.5|28.8±3.1|29.6±2.7|20.8±0.7|19.2±0.7|
> |Swap-VAE|47.2±2.6|32.8±3.1|26.4±0.9|33.6±3.3|28.8±1.3|
> |CEBRA|36.0±3.0|27.2±2.1|29.6±1.4|32.8±1.3|**36.0±1.1**|
> |**TiDeSPL-VAE**|**58.4±2.7**|**35.2±3.5**|**36.8±2.4**|**42.4±2.1**|32.0±1.6|
>
> Table R1: The decoding results (\%) on five new scenes.
>
> |Model|Mouse 1|Mouse 2|Mouse 3|Mouse 4|Mouse 5|
> |-|:-:|:-:|:-:|:-:|:-:|
> |LFADS|49.6±3.2|49.2±3.9|42.0±0.6|47.6±1.3|43.6±2.2|
> |pi-VAE|56.0±2.2|55.6±3.3|**53.6±2.9**|47.6±1.5|37.2±0.9|
> |Swap-VAE|46.4±5.2|38.4±2.1|29.6±1.4|44.8±1.5|38.0±1.3|
> |CEBRA|30.4±1.5|26.4±1.8|27.6±2.5|29.6±2.1|22.4±3.0|
> |**TiDeSPL-VAE**|**68.4±2.0**|**56.8±1.8**|51.6±2.5|**51.6±0.9**|**45.6±1.8**|
>
> Table R2: The decoding results (\%) on all ten scenes.
>
> **3. Some references are missing.**
>
> Thank you for your suggestion. These references are a good complement to our related work. The first [2] combined a deep network for stimulus-driven activity and a flow-based generative model to capture the stimulus-conditioned variability, providing new insights into latent state construction. The second [3] introduced structured state-space layers into the autoencoder to project discrete high-dimensional spiking data into continuous time-aligned latent states, simultaneously achieving inference of low-dimensional latent states and realistic conditional generation of neural spiking data. While the former uses visual stimuli to predict the distribution of neural population responses and the latter aims to reconstruct realistic neural activity, our work focuses on building latent representations from neural activity and decoding visual stimuli. We will add these discussions in the revision.
>
> **4. RNNs are often tricky to train.**
>
> We have found that our model is prone to overfitting, especially when dealing with long sequential data. We have largely mitigated this problem by adjusting the learning rate and applying the techniques of learning rate decay and early stopping, which enables our model to achieve the best performance.
>
> **5. Did you train a new separate model per mouse?**
>
> Since each mouse is implanted with Neuropixel probes separately in biological experiments, there is no guarantee that the neural sites recorded from each mouse are aligned, and the number of recorded neurons is different. Therefore, we train separate models for each mouse.
>
> **6. About Figure 3.**
>
> We think this is similarly related to the large variability in neural activity across subjects and trials (as discussed in Response 2). Specifically, on some trials, Mouse 1's neural activity is easily distinguishable when it is focusing its attention on the visual stimulus when , but on other trials, its neural activity may be dominated by its own state rather than driven by stimuli. As shown in Figure 7 in the Appendix, our model makes a good separation between stimulus-based clusters, especially for Mouse 2, suggesting that analysis of subject-to-subject and trial-to-trial variability is a meaningful direction for future investigation.

---

> ### Author Response · Authors · 2024-11-22
>
> **7. How exactly the train/test splits are done?**
>
> As described in lines 302-308, each visual stimulus is presented for multiple trials, and we divide all trials into training, validation, and test sets, rather than dividing them with respect to the time. In training, positive samples are selected only within the same trial and do not involve trials in the validation and test sets. Therefore, we ensure that none of the validation/test responses were shown during training.
>
> [1] Tyler D Marks and Michael J Goard. Stimulus-dependent representational drift in primary visual cortex. Nature communications, 2021.
>
> [2] Mohammad Bashiri, et al. A flow-based latent state generative model of neural population responses to natural images. NeurIPS, 2021.
>
> [3] Jaivardhan Kapoor, et al. Latent Diffusion for Neural Spiking Data. arXiv, 2024.

---

> ### Comment · Reviewer_MX7e · 2024-11-23
>
> Thanks a lot for clarifications!
>
> One more question about train-val-test split. Did I got correctly that responses for the repeats of the same stimuli can appear in both train and test? (If I show image A at time 10 and 20, the responses for the first presentation can go to test and for the second - to train?)
>
> And as you train a model per animal, have you thought about some adapter layer to unify the dimensions and then keep the shared encoder-decoder structure, reported in the paper? What are the limitations of this approach? Otherwise, the latent states between mice are not necessary easily comparable due to the potential misalignment between the models. (See  [1] for more misalignment  details)
>
> [1] Moschella, L., Maiorca, V., Fumero, M., Norelli, A., Locatello, F., & Rodolà, E. (2022). Relative representations enable zero-shot latent space communication. arXiv preprint arXiv:2209.15430.

---

> > ### Author Response · Authors · 2024-11-25
> >
> > Thank you for providing the constructive feedback.
> >
> > **1. About train-val-test split.**
> >
> > We did apply neural activity for the repeats of the same stimuli in both training and test datasets, but without overlap. Considering the significant trial-to-trial variability of neural activity, this approach is common in the neuroscience field [1-3]. In other words, the neural activity for the repeats of the same stimuli is usually treated as different trials of a condition, like different samples of a category in the computer vision field. What's more, in the biological experiment, 118 image stimuli were presented in a random order, and the probability of the same image stimulus being presented next to itself is low, thus the causal correlation between them is scarce.
> >
> > **2. Some adapter layer to unify the dimensions.**
> >
> > We think this is a very insightful suggestion. Indeed, LFADS has attempted to use one-layer read-in and read-out to unify the dimensions and jointly train multi-session neural data, but this requires frequent supervised recalibration when dealing with new data [4]. Furthermore, some work has applied advanced ML techniques, such as diffusion models and transfer learning, to align latent representations of neural data from different sessions and animals [5-6]. However, most work has focused on motor brain regions, where the underlying dynamics identified from different data on the same task are similar [7]. By comparison, neural variability and statistical heterogeneities between animals and sessions can be very large in visual cortical regions (as also observed in the results of our work), which may increase the difficulty of alignment and even compromise the quality of latent representations extracted from a single animal. Nevertheless, we agree that the misalignment exists and should be addressed for more effective analyses of the latent states between mice. We will explore alignment methods for the visual cortex in the future.
> >
> > [1] Alessandro Scaglione, et al. Trial-to-trial variability in the responses of neurons carries information about stimulus location in the rat whisker thalamus. PNAS, 2011.
> >
> > [2] Ran Liu, et al. Drop, swap, and generate: A self-supervised approach for generating neural activity. NeurIPS, 2021.
> >
> > [3] Steffen Schneider, Jin Hwa Lee, and MackenzieWeygandt Mathis. Learnable latent embeddings for joint behavioral and neural analysis. Nature, 2023.
> >
> > [4] Chethan Pandarinath, et al. Inferring single-trial neural population dynamics using sequential auto-encoders. Nature methods, 2018.
> >
> > [5] Brianna M. Karpowicz, et al. Stabilizing brain-computer interfaces through alignment of latent dynamics. bioRxiv, 2022.
> >
> > [6] YuleWang, et al. Extraction and Recovery of Spatio-Temporal Structure in Latent Dynamics Alignment with Diffusion Models. NeurIPS, 2023.
> >
> > [7] Ayesha Vermani, et al. Leveraging Generative Models for Unsupervised Alignment of Neural Time Series Data. ICLR, 2024.

---

### Official Review · Reviewer_35tP · 2024-11-01

**Soundness:** 3
**Presentation:** 2
**Contribution:** 1
**Rating:** 6
**Confidence:** 3

**Summary:**

The paper introduces a new model, the TiDeSPL-VAE, to decode mice's neural activity during passive viewing of natural images and movies.
The TiDeSPL-VAE is a recurrent auto-encoder-like model composed of two latent spaces, each updated based on the current input and the recurrent state of the previous input. The model is also trained using a combination of losses: reconstruction, KL, and contrastive.
All in all, the authors demonstrate that the TiDeSPL-VAE outperforms numerous baselines in reconstructing spiking activity on synthetic and naturalistic datasets.

**Strengths:**

The authors have been rigorous in the research methodology, comparing their model against a wide variety of baselines and testing the ability of these models on different datasets, from synthetic to more naturalistic settings -- and a complementary ablation detailing the critical components of the model.

The authors also attempt to solve a novel problem by decoding visual data from mice instead of the widely studied mice's motor system.

**Weaknesses:**

**Major Weakness**: The authors insist on the novelty of their work as it deals with visual data instead of motor data from mice. However, the format of the data -- spiking activity -- is treated similarly, whether it is from the motor or the visual cortex. The model still takes the spiking activity as input and outputs the reconstructed spiking activity. To take advantage of the visual modality, it would be interesting to evaluate whether it is possible to decode the input stimuli (image or movie) or to relate the model better with neuroscientific findings about the mouse visual cortex. As it is presented now, the model could also be employed on motor data and perform similarly well. The use of visual data should be better exploited to highlight the scientific advances of this article more than the methodological part.

**Minor Weaknesses**:
- The writing should be slightly improved to make the paper easier to read. A lot of sentences are not very clear, and others have grammatical flaws. For example, L269 "... and others are even difficult to split four clusters ..."; L482 "contrast learning"; ...

**Questions:**

- Why are the reconstruction and KL losses computed on the positive samples as well as the other samples? Isn't there an incentive already with the contrastive loss to emphasize positive samples? Can the authors include in the ablation study the training with the losses on all samples only to evaluate the impact of these terms?

- L259, the authors mention that the two synthetic datasets consider "different properties of visual neural activity". What are these properties?

- It looks like the authors train a model for each mouse and treat each mouse as a single dataset. How different would the results be if the authors tried training on all the mice? Or training on some of them and testing on the left-out individuals?

- Can the authors include in appendix the number of samples (train, val and test) for each dataset? It looks like, with all the exclusion criteria, the dataset size ends up being very small. Can the authors confirm that the models are not overfitting on data of a single animal?

- Can the authors also include a table indicating the number of parameters of each model in the Appendix to confirm that the comparisons are fair?

- The ablation study highlights the importance of the contrastive loss and swap operation. Without these objectives, the model does not outperform most of the baselines. But the model has other characteristics that the baselines do not have such as the recurrence due to the two RNNs and two latent spaces. To ensure that these additions are important, can the authors include in the ablation study a non-recurrent version of the model (i.e. setting the number of timesteps of the RNNs to 1), and a version without each of the latent spaces?
Additionally, would it be possible to train one of the best baselines with the contrastive loss and the swap operation?

---

> ### Author Response · Authors · 2024-11-22
>
> We thank the reviewer for the notable and perceptive comments. We will do our best to address the comments and provide detailed responses point by point, below.
>
> **1. About the novelty.**
>
> We agree that it is important to take advantage of the visual modality when dealing with visual neural data. Indeed, our work focuses on **revealing the intrinsic correlation between neural activity and visual stimuli**. As described in lines 243-246, we **decode visual stimuli using low-dimensional latent representations** transformed from neural activity and assess the clarity of latent temporal structures to **explain neural dynamics under different types of visual stimuli**. In Tables 2 and 3 of the main text, we report the decoding performance for natural scene/movie frame classification, showing that our model outperforms other baselines. In Figures 4 and 5, we exhibit the latent structures and analyzed their characters corresponding to visual stimuli. Ablation studies are also based on the decoding performance of visual stimuli.
>
> Our lack of emphasis on the evaluation metric on neural datasets may have led to your misunderstanding. We will describe the evaluation in more detail in the revision.
>
> **2. About the writing.**
>
> We apologize for the unclear expression or grammatical flaws in our manuscript. We will modify the sentence in line 269 to "others even struggle to split four clusters" and the word in line 482 to "contrastive learning". What's more, we will re-check and improve our writing.
>
> **3. Why are the reconstruction and KL losses computed on the positive samples as well as the other samples?**
>
> These loss terms are motivated by the emphasis on the positive samples. As suggested by the reviewer, we perform an ablation study to remove these losses computed on the positive samples and report the results in Tables R1 and R2. The results show that these loss terms contribute less to performance, suggesting that the contrastive loss may be sufficient to emphasize the positive samples. We will consider removing these loss terms to speed up model training in the future.
>
> |Model|Mouse 1|Mouse 2|Mouse 3|Mouse 4|Mouse 5|
> |-|:-:|:-:|:-:|:-:|:-:|
> |**TiDeSPL-VAE**|**96.4±1.1**|**74.8±2.0**|**74.8±1.7**|**78.8±2.9**|**67.6±2.1**|
> |Without losses on the positive samples|96.0±1.7|71.2±1.8|74.4±1.4|77.2±1.4|65.6±1.9|
>
> Table R1: The ablation results (\%) for losses on the positive samples in natural scene decoding experiments.
>
> |Model|Mouse 1|Mouse 2|Mouse 3|Mouse 4|Mouse 5|
> |-|:-:|:-:|:-:|:-:|:-:|
> |**TiDeSPL-VAE**|**13.88±0.19**|**65.38±0.36**|**59.88±0.72**|**54.33±0.54**|**30.18±0.40**|
> |Without losses on the positive samples|12.80±0.19|64.18±0.42|58.70±0.53|54.00±0.39|29.97±0.50|
>
> Table R2: The ablation results (\%) for losses on the positive samples in natural movie frame decoding experiments.
>
> **4. What are these properties in the synthetic datasets?**
>
> As described in lines 259-261, the first property is the different categories of visual stimuli that visual neural activity corresponds to. Consequently, the non-temporal dataset is generated from several sets of labels (four clusters). The second property is the time dependence of neural activity. Taking this into account, the temporal dataset is constructed using the Lorenz system.

---

> ### Author Response · Authors · 2024-11-22
>
> **5. Training on all the mice? Or training on some of them and testing on the left-out individuals?**
>
> In biological experiments, each mouse is implanted with Neuropixel probes separately, so there is no guarantee that the neural sites recorded from each mouse are aligned, and the number of recorded neurons is different (Table 6 in the Appendix). Therefore, we treat each mouse as a single dataset. As suggested by the reviewer, we sample neurons from all mice evenly to form a dataset (All Mice dataset) with a similar number of neurons to the single mouse dataset. The results in Tables R3 and R4 show that our model performs better than other models on this dataset, but the performance is overall lower than that on the single mouse dataset, suggesting variability in neural activity to the same visual stimuli across mice.
>
> ||LFADS|pi-VAE|Swap-VAE|CEBRA|TiDeSPL-VAE|
> |-|:-:|:-:|:-:|:-:|:-:|
> |**All Mice**|41.6±4.0|44.0±3.8|34.4±4.9|46.4±3.7|**49.6±2.4**|
>
> Table R3: The results (\%) on the All Mice dataset in natural scene decoding experiments.
>
> ||LFADS|pi-VAE|Swap-VAE|CEBRA|TiDeSPL-VAE|
> |-|:-:|:-:|:-:|:-:|:-:|
> |**All Mice**|16.36±1.02|22.44±0.53|22.07±0.28|22.98±0.76|**28.07±1.01**|
>
> Table R4: The results (\%) on the All Mice dataset in natural movie frame decoding experiments.
>
> In addition, we sample the same number of neurons in each mouse. We then train our model on one mouse and test it on the other. As shown in Tables R5 and R6, for the same mouse in the test, there is a significant drop in performance for models trained on other mice. This may be due to the fact that it is hard to align the recorded neurons and there is a large variation in response patterns across mice.
>
> |Train \ Test|Mouse 1|Mouse 2|Mouse 3|Mouse 4|Mouse 5|
> |-|:-:|:-:|:-:|:-:|:-:|
> |**Mouse 1**|**88.8±2.1**|48.0±2.2|59.2±3.2|60.0±3.0|50.4±2.6|
> |**Mouse 2**|54.0±3.9|**70.0±3.5**|51.6±3.7|64.0±3.1|48.0±2.9|
> |**Mouse 3**|56.0±2.9|48.8±3.4|**70.0±2.7**|58.0±3.4|50.4±2.1|
> |**Mouse 4**|63.2±4.0|53.2±3.3|51.6±1.7|**75.2±1.9**|56.0±2.9|
> |**Mouse 5**|64.4±2.3|42.8±1.6|42.4±2.7|54.4±3.1|**69.6±3.1**|
>
> Table R5: The results (\%) of our model trained and tested on different mouse datasets in natural scene decoding experiments.
>
> |Train \ Test|Mouse 1|Mouse 2|Mouse 3|Mouse 4|Mouse 5|
> |-|:-:|:-:|:-:|:-:|:-:|
> |**Mouse 1**|**11.42±0.30**|25.87±0.36|20.76±0.62|23.56±0.45|12.83±0.58|
> |**Mouse 2**|8.60±0.23|**60.41±0.62**|17.47±0.50|18.91±0.54|10.79±0.29|
> |**Mouse 3**|8.32±0.13|19.82±0.88|**52.50±0.84**|19.53±0.41|11.43±0.32|
> |**Mouse 4**|8.42±0.35|20.24±0.63|18.08±0.51|**51.18±0.64**|11.26±0.31|
> |**Mouse 5**|8.34±0.29|21.00±0.51|18.60±0.35|19.51±0.54|**28.10±0.50**|
>
> Table R6: The results (\%) of our model trained and tested on different mouse datasets in natural movie frame decoding experiments.
>
> **6. The number of samples for each dataset.**
>
> We report the number of samples for each dataset in Table R7, where the size of the mouse dataset under natural scene stimuli is somewhat small. In training, we use a small learning rate and early stopping to ensure that the models are not overfitting.
>
> |Datasets|Train|Validation|Test|
> |-|:-:|:-:|:-:|
> |**synthetic non-temporal dataset**|12800|1600|1600|
> |**synthetic temporal dataset**|80000|10000|10000|
> |**mouse dataset under natural scene stimuli**|5000|625|625|
> |**mouse dataset under natural movie stimuli**|28800|3600|3600|
>
> Table R7: The number of samples for each dataset.
>
> **7. The number of parameters of each model.**
>
> The number of model parameters is roughly proportional to the number of input neurons, so here we only report the number of parameters for the Mouse 1 dataset (Table R8), and we will include other results in the Appendix. Due to incorporating the recurrent modules, our model has slightly more parameters than some models and is comparable to CEBRA.
>
> ||beta-VAE|LFADS|pi-VAE|Swap-VAE|CEBRA|TiDeSPL-VAE|
> |-|:-:|:-:|:-:|:-:|:-:|:-:|
> |**Number of Parameters**|0.39M|0.45M|0.49M|0.38M|0.71M|0.68M|
>
> Table R8: The number of model parameters for the Mouse 1 dataset.

---

> ### Author Response · Authors · 2024-11-22
>
> **8. The ablation studies on the RNNs and latent spaces.**
>
> As suggested by the reviewer, we conduct ablation studies on the RNNs and latent spaces. As shown in Tables R9 and R10, the non-recurrent version and the two versions without content/style latent space almost all perform worse than our original model, suggesting that the RNNs and the two latent spaces play a critical role.
>
> |Model|Mouse 1|Mouse 2|Mouse 3|Mouse 4|Mouse 5|
> |-|:-:|:-:|:-:|:-:|:-:|
> |**TiDeSPL-VAE**|**96.4±1.1**|**74.8±2.0**|**74.8±1.7**|**78.8±2.9**|**67.6±2.1**|
> |Non-recurrent|82.0±2.2|52.0±3.1|56.4±2.7|68.0±2.6|59.6±4.1|
> |Only with content|74.0±2.0|49.6±2.8|53.2±2.3|68.8±3.8|56.0±3.1|
> |Only with style|60.0±3.0|32.4±2.6|44.4±2.6|47.2±1.7|44.8±2.8|
>
> Table R9: The ablation results (\%) for the RNNs and latent spaces in natural scene decoding experiments.
>
> |Model|Mouse 1|Mouse 2|Mouse 3|Mouse 4|Mouse 5|
> |-|:-:|:-:|:-:|:-:|:-:|
> |**TiDeSPL-VAE**|**13.88±0.19**|**65.38±0.36**|**59.88±0.72**|**54.33±0.54**|30.18±0.40|
> |Non-recurrent|11.14±0.30|53.26±0.48|48.31±0.47|43.81±0.40|22.98±0.44|
> |Only with content|12.16±0.24|63.33±0.29|58.64±0.37|52.11±0.39|**30.24±0.54**|
> |Only with style|7.57±0.24|11.73±0.43|12.86±0.35|16.10±0.33|9.89±0.22|
>
> Table R10: The ablation results (\%) for the RNNs and latent spaces in natural movie frame decoding experiments.
>
> Besides, Swap-VAE, as one of the best models we compare in our work, includes the swap operation and uses an alignment loss computed on positive sample pairs which plays a similar role to the contrastive loss. Its performance is slightly worse than or comparable to the non-recurrent version of our model, demonstrating the importance of RNNs in introducing time dependence.

---

> ### Author Response · Authors · 2024-11-25
> **Looking forward to reviewer's feedback**
>
> Dear Reviewer 35tP,
>
> Thank you for your valuable time and constructive comments. As the discussion period will end soon, we hope to discuss whether your concerns have been addressed and would appreciate it if you could reconsider the rating. If you have any further questions, we would be happy to discuss them with you and do our best to address them.
>
> Best,
>
> The authors

---

> > ### Comment · Reviewer_35tP · 2024-11-25
> >
> > I would like to thank the authors for replying to the review, but I would like to point out that I do not see any of these results appearing in the manuscript. Therefore, I wonder whether the authors planned to include these additional results in the manuscript.
> >
> > However, my main concerns remain unresolved: even though the neural activity reflects the properties of visual stimuli, the model still reconstructs the neural activity and not the image or the movie (as illustrated in Fig. 1). Classifying the stimuli is akin to a clustering problem that can be done with any modality and is still performed on the neural activity.
> >
> > Additionally, SwapVAE is the most competitive model to TiDeSPL-VAE despite having almost half of the parameter count of TiDeSPL-VAE. A control experiment showcasing the performance of SwapVAE with the same number of parameters as TiDeSPL-VAE would help convince me whether TiDeSPL-VAE has additional abilities compared to SwapVAE. Otherwise, SwapVAE strikes me as a better choice for solving the task as it is simpler and faster than the TiDeSPL-VAE.
> >
> > For these reasons, I will keep my score unchanged.

---

> ### Author Response · Authors · 2024-11-28
>
> We are very sorry that we didn't update our manuscript timely. We have included all the clarifications and additional results in our revision according to the reviewers' suggestions. Furthermore, we would like to address your remaining concerns.
>
> **1. About reconstructing neural activity and classifying stimuli.**
>
> First, we would like to emphasize that our work focuses on constructing low-dimensional high-quality latent representations from neural activity to reveal the intrinsic correlation between neural activity and visual stimuli. Second, VAE-based latent variable models (LVMs) aim to build latent representations, while reconstructing neural activity is not the research objective, but rather an objective of model training. Third, the use of latent representations for classification tasks is an important quantification of the quality of latent representations, which is a common approach in studies oriented to the motor cortex [1-3] and the visual cortex [4-5]. Last but not least, directly reconstructing visual stimuli is another research objective and usually requires neural activity with much larger datasets, such as fMRI [6]. In conclusion, our work follows the common analytical approach in the field, and our model achieves the highest performance on widely used evaluation methods.
>
> **2. A small version of TiDeSPL-VAE.**
>
> As suggested by the reviewer, we build a small version of our model (TiDeSPL-VAE-small) with fewer trainable parameters than Swap-VAE. Since the introduction of the recurrent module inevitably increases the number of parameters, we reduce the parameters of the encoder and decoder to achieve this (see Appendix E of the revision for the number of parameters). As shown in Tables R1 and R2, the small version of our model still outperforms Swap-VAE in most cases, suggesting that the performance improvement mainly stems from the design of our model rather than the larger number of parameters. We have included these results in Table 2 and 3 of our revised manuscript.
>
> |Model|Mouse 1|Mouse 2|Mouse 3|Mouse 4|Mouse 5|
> |-|:-:|:-:|:-:|:-:|:-:|
> |Swap-VAE|86.0±1.6|70.8±2.0|54.4±1.5|67.2±2.9|59.6±3.5|
> |**TiDeSPL-VAE-small**|90.0±2.0|65.2±3.2|71.2±2.2|74.4±2.7|**70.8±2.0**|
> |**TiDeSPL-VAE**|**96.4±1.1**|**74.8±2.0**|**74.8±1.7**|**78.8±2.9**|67.6±2.1|
>
> Table R1: The decoding results (\%) in natural scene decoding experiments.
>
> |Model|Mouse 1|Mouse 2|Mouse 3|Mouse 4|Mouse 5|
> |-|:-:|:-:|:-:|:-:|:-:|
> |Swap-VAE|11.09±0.25|44.99±0.76|36.37±1.53|37.68±1.14|19.14±0.63|
> |**TiDeSPL-VAE-small**|12.26±0.30|63.30±0.34|57.57±0.39|53.46±0.58|28.70±0.42|
> |**TiDeSPL-VAE**|**13.88±0.19**|**65.38±0.36**|**59.88±0.72**|**54.33±0.54**|**30.18±0.40**|
>
> Table R2: The decoding results (\%) in natural movie frame decoding experiments.
>
> [1] Chethan Pandarinath, et al. Inferring single-trial neural population dynamics using sequential auto-encoders. Nature methods, 2018.
>
> [2] Ding Zhou and Xue-Xin Wei. Learning identifiable and interpretable latent models of highdimensional neural activity using pi-vae. NeurIPS, 2020.
>
> [3] Ran Liu, et al. Drop, swap, and generate: A self-supervised approach for generating neural activity. NeurIPS, 2021.
>
> [4] Steffen Schneider, Jin Hwa Lee, and MackenzieWeygandt Mathis. Learnable latent embeddings for joint behavioral and neural analysis. Nature, 2023.
>
> [5] Kendrick N Kay, et al. Identifying natural images from human brain activity. Nature, 2008.
>
> [6] Jiaxuan Chen, et al. Rethinking Visual Reconstruction: Experience-Based Content Completion Guided by Visual Cues. ICML, 2023.

---

> > ### Comment · Reviewer_35tP · 2024-12-02
> >
> > Thank you to the authors for running additional experiments and addressing some of my concerns.
> > While I'm still not convinced of the novelty of the work regarding visual stimuli, I acknowledge the work done around the method proposed by the authors. For this reason, I raise my score to 6.

---

> ### Author Response · Authors · 2024-12-02
>
> Thank you for recognizing our work. More importantly, we sincerely appreciate your valuable comments and suggestions that have helped us improve our work.

---

### Official Review · Reviewer_XzNA · 2024-11-03

**Soundness:** 2
**Presentation:** 1
**Contribution:** 2
**Rating:** 3
**Confidence:** 4

**Summary:**

This paper proposes TiDeSPL-VAE, a sequential latent variable model that aims to jointly model neural and visual inputs. The model uses separate embeddings to represent visual features and internal states. However, there are significant issues in motivation and clarity that hinder evaluating its contributions in its current form.

**Strengths:**
- The idea of separating visual features from internal states could be beneficial for neuroscience and machine learning, with potential applications in visual decoding.

**Concerns:**
1. **Motivation and Related Work**: The study’s motivation is weak and lacks clarity. Statements on motor encoding and temporal relationships lack precision, and relevant prior work on task-relevant/irrelevant models is omitted. Checking for grammar would also be good to do.
2. **Benchmarking and Fairness**: No clear details on benchmarking models, hyperparameter choices, or treatment of time dependencies are provided. Models like CEBRA and LFADS are compared inadequately due to differences in architecture and training parameters. Additionally, benchmarks on common datasets like the Allen Brain Natural Movie 1 or Sensorium Challenge data are missing, limiting reproducibility/interpretability.
3. **Scientific  Impact**: It remains unclear if the model’s components contribute unique insights, as it largely combines established techniques (VAEs, contrastive learning) without a compelling scientific rationale. The model complexity, evidenced in ablations, appears to drive performance rather than underlying innovations in latent space separation.

**Recommendation:**
While TiDeSPL-VAE may hold promise, the paper’s current form lacks clarity and adequate benchmarking, raising concerns about model novelty and effectiveness. Addressing these issues is essential for it to be a meaningful addition to the field.

**Strengths:**

Anonymous et al. present a method for joint modeling of neural and visual inputs called TiDeSPL-VAE. The proposed method leverages a sequential latent variable model (LVM) to learn two separate embeddings - one to represent the visual latent features and one to capture the animals' “internal state.” As such, building robust and interpretable latent variable models is of great interest to fundamental neuroscience, translational work (restoring vision), and in machine learning. However, due to missing details and critical baselines (see below), it is not clear the originality or value of the solution they present.

**Weaknesses:**

I find the study poorly motivated with statements such as *“Most work has focused on analyzing motor neural activity that controls clear behavioral traces and has modeled neural temporal relationships in a way that does not conform to natural reality.”* - what is natural reality?
Also, note that actual realistic motor behavior encoding-decoding is an unsolved problem.

Or equally worrisome: *“even though how the visual system encoded input to recognize objects is a primary topic(DiCarlo et al.,2012), and decoding visual neural activity visual stimuli is a challenging research highlight in the neuroscience community (Kay et al.,2008; Wen et al.,2018; Duetal.,2023). Furthermore, existing LVM treat temporal relationships unnaturally (Pandarinathetal., 2018; Schneider et al., 2023) or even don't consider time dependence (Zhou & Wei, 2020; Palmerston & Chan, 2021). Given that visual neural activity has strict antecedent time dependence, these models may struggle to build high-quality latent representations.”* - as a trained neuroscientist and ML expert, I can confidently say this is not neurally accurate. Also, how does LFADS or CEBRA treat “temporal relationships” unnaturally? LFADS uses a dynamical system and CEBRA can leverage the natural time inherent in the neural code, which is a tunable hyperparameter.

**Related Works/Motivation**

- Why not motivate this study by just stating what you do – you want to see if separating visual features from internal states could lead to better LVMs for understanding (decoding) visual systems.

- You also miss the related work on task-relevant and task-irrelevant models, such as [1](https://scholar.google.com/citations?view_op=view_citation&hl=en&user=csGAeKgAAAAJ&sortby=pubdate&citation_for_view=csGAeKgAAAAJ:WbkHhVStYXYC) and [2](https://pubmed.ncbi.nlm.nih.gov/37398400/), which should be benchmarked.

- You also claim, of the models you benchmark: *“None of them build latent representations progressively along the chronological order.”* (L253) - what does this mean?

**Benchmarking Issues**

- In the main text or Appendix, there are NO details on how you benchmarked the other models, aside from *“In this experiment, we set all models to have 128-dimensional latent variables and train them for 5,000 iterations”*. How is this a fair comparison? Did you not try other dimensions and training losses until convergence? How did you treat time in CEBRA or LFADS, as you can set different offsets - you only mentioned once: *“5 time points for natural scenes and for natural movie”*, but this does not tell me about the offset time in the model?

- Moreover, the complexity of the different models - B-VAE, LFADS, CEBRA are extremely different. For example, the work from Schneier et al. 2023 (CEBRA)  uses only a 5-layer MLP/GeLU, as the innovation is in the generalized contrastive loss - the point was you can get great performance with a simple model; to fairly benchmark this, then you should use the same RNN as the base model (vs. their MLP). All models you test have many hyperparameters and a full analysis of the parameters you tested is absolutely critical to show.
- You additionally need to compare model complexity and ideally, run-time estimations.

- You use the Allen Brain Observatory Dataset because you allude that this is the data that Schneider et al. 2023 use, which, I agree, would be ideal. However, you do not benchmark on Natural Movie 1 in the same way; rather, your main benchmark is Natural Images – notably, devoid of time in the stimulus space. It is strange to me that you don’t benchmark on any data - synthetic or otherwise - that anyone else has used in ML to benchmark on. It raises the concern that you are unfairly enhancing the performance of your approach compared to letting the original authors run their code on common data. See my points above.

- Thus, my strong recommendation is that you should also test your approach on “motor tasks” on standardized benchmarks, such as the Neural Latent Benchmark, as well as established vision benchmarks, such as the [Sensorium Challenge](https://www.sensorium-competition.net/) for natural images. At the very least, the Allen Brain Natural Movie 1 data as it was presented in CEBRA where you can add their numbers to a Table to directly compare; here you don’t reach the same decoding performance and you seemingly pick the neurons differently (they report close to 90%, which here your overall is much lower (65%), again making it hard to compare.

- Scientifically, I am also missing how the separate embeddings perform in tasks; by separating vision from task-irrelevant information, do you see a more interpretable latent space?

- Moreover, it is unclear to me how this is not just an A+B solution; you leverage VAEs, and swaps, and contrastive learning, I don't inherently see the novelty (but this does not affect my score, novelty is a construct anyhow), but I would like to better understand the scientific rationale for the design choice.

**Summary**

I believe your approach could be useful for the field, but in the paper's current form:

1. Lacks the proper motion and clarity in grammar/thought.
2. It is impossible to know if you did a fair comparison due to lacking details, and common benchmarks, and the missing required benchmarking of alternative methods.
3. Missing task-relevant and task-irrelevant baselines.
4. Your method is more complicated than those you benchmark, and your ablation studies hint at this as well. Comparing Mouse 2 in Table 3 you reach 65.38%, while the second best (much simpler) method reaches 52.76% decoding performance. Now, in Table 4, if you remove anything, you are well below the 52.76%, suggesting that the complexity of your model is driving performance. Just as another test, if you drop in an MLP (as you did from GRU→ LSTM), do you still outperform other models? I would suspect not.

**Questions:**

Please see above weaknesses.

---

> ### Author Response · Authors · 2024-11-22
>
> We thank the reviewer for the clear and pointed comments. We will try our best to address the reviewer's concerns and answer the questions in the following.
>
> **1. About the Motivation.**
>
> We would like to elaborate on some of the statements and clarify our motivation. First, what is natural reality and antecedent time dependence? Neural activity at the current time evolves only from past neural activity but does not depend on future neural activity. Therefore, **modeling time dependence and constructing latent representations in a natural way should use only current and past neural activity**.
>
> Based on this motivation, we state that LFADS and CEBRA treat temporal relationships unnaturally since they both use future information to construct latent representations of the current time. As discussed in lines 251-254, although LFADS uses dynamical systems at the generator/decoder, **its encoder uses bidirectional RNNs**. Consequently, its decoder constructs latent variables at all times using future information, and even includes temporal relationships that evolve from the future to the past. As for CEBRA, **it takes the surrounding time points centered on the current point** and uses time convolution to encode temporal features, resulting in the inclusion of future information and the absence of chronological relationships.
>
> In contrast, our work uses unidirectional RNNs to **construct latent representations that depend only on current and past neural activity**, which is an equally important contribution as our design for splitting latent variables. Therefore, we state that **our model builds latent representations progressively along the chronological order**. However, we agree that realistic motor behavior encoding-decoding is an unsolved problem and we will check the grammar in our manuscript.
>
> **2. About the related works.**
>
> Thank you for your suggestion. We will include discussions about task-relevant and task-irrelevant models to our related work. Besides, we want to address that we have benchmarked several task/behavior-relevant models, such as pi-VAE and CEBRA, as well as task-irrelevant models, such as LFADS and Swap-VAE. Unfortunately, our work focuses on spiking activity recorded from the mouse visual cortex under natural scene/movie stimuli, thus we did not notice the study in [2], which provides an accurate and efficient method for learning multiscale dynamical models for multimodal spike-LFP recordings of a monkey performing a reach-and-grasp movement task. (BTW, the link to ref [1] can not be accessed). We will evaluate this model in our experiments in the future.
>
> **3. About the training settings of the other models.**
>
> As described in lines 318-319 and Appendix E, we have provided detailed experiment setups for baseline models. For the latent variable dimension, we choose 128 for fair comparisons, because **it was used in Swap-VAE and CEBRA and produced the highest task performance**. We have also conducted ablation studies on the dimension (Figure 5), showing that the model performance saturates gradually as the dimension of latent variables increases and suggesting that it is sufficient to choose a dimension at reasonable intervals. For the training iteration, we adjust it based on the size of the dataset and ensure that **the losses for our model and baseline models all converge**.
>
> The input time sequence processing of LFADS and CEBRA has been presented in Appendix E. In each experiment, **the length of their input samples is consistent with our model**, 5 time points for natural scenes (lines 313-314) and 4 time points for natural movie (lines 425-426). Besides, we have also provided the offset time, ±3 for natural scenes (lines 314-315) and ±2 for natural movie (lines 425-426).

---

> > ### Comment · Reviewer_XzNA · 2024-11-26
> >
> > Thanks for your clarifications; I will note, that, however, this seems incorrect: "As for CEBRA, it takes the surrounding time points centered on the current point". In their paper they show no surrounding timepoints when they benchmark tSNE, UMAP, LFADS (offsetmodel1) - adding timepoints is optional, and my understanding is that because neural activity is often delayed to the behavior, -- the motor cortex signals are before EMG/video data, and their is preparatory activity such that you want to take a small time video of neural data. Nonetheless, my point is that calling other algorithms "unnatural" doesn't aid in your papers presentation.

---

> ### Author Response · Authors · 2024-11-22
>
> **4. About the complexity.**
>
> In terms of model structure and loss function design, we admit that our model is more complex. However, each component has a clear conceptual interpretation and motivation (as described in Section 3). For example, we choose GRU to address the lack of long-time dependence inherent in RNNs, which is also used in [LFADS](https://github.com/arsedler9/lfads-torch). In addition, we have performed ablation studies replacing GRU with vanilla RNN (Tables 4 and 7 in the main text), and **the results show that even with RNN, our model still achieves the highest performance in most cases**. For the hyperparameters, we have searched and adjusted them for all models to ensure that their losses converge after training and they achieve the best performance.
>
> In terms of model trainable parameters, we report the number of parameters for the Mouse 1 dataset for example (Table R1), as the number of model parameters is roughly proportional to the number of input neurons. Due to incorporating the recurrent modules, **our model has slightly more parameters than some models and is comparable to CEBRA**. In terms of running time, since our model only uses current and past information, it can only process neural data serially, and the running time of our model is indeed longer compared to models without time dependence and CEBRA, which uses time convolution that can be operated in parallel.
>
> ||beta-VAE|LFADS|pi-VAE|Swap-VAE|CEBRA|TiDeSPL-VAE|
> |-|:-:|:-:|:-:|:-:|:-:|:-:|
> |**Number of Parameters**|0.39M|0.45M|0.49M|0.38M|0.71M|0.68M|
>
> Table R1: The number of model parameters for the Mouse 1 dataset.

---

> ### Author Response · Authors · 2024-11-22
>
> **5. About the benchmark datasets.**
>
> First, we have not only benchmarked on natural scenes but also on the natural movie. **The decoding score metric follows that proposed by CEBRA (line 431)**. Second, as described in lines 259-264, the synthetic data follows some previous work. The non-temporal dataset was used in Swap-VAE and the temporal dataset was used in LFADS.
>
> However, we agree that we should test our model on more existing benchmarks, as suggested by the reviewer. First, we evaluate our model on the [Allen Brain Natural Movie 1](https://cebra.ai/docs/demo_notebooks/Demo_Allen.html) presented in CEBRA. In contrast to our use of the neural activity of each mouse independently as a dataset, CEBRA sampled neurons from the same brain regions in all mice to form the dataset. The experiment setup and model training fully follow the demo provided by CEBRA and the description in their paper. As shown in Table R2, **our model outperforms CEBRA on datasets with different numbers of neurons sampled**. However, we observe that our model's advantage decreases with more input neurons, which may be due to the fact that CEBRA's encoder has more layers than ours (we have only two), resulting in a lack of feature encoding ability in the neuron dimension of our model. Besides, these results are obtained in the case of a sample with 4 time steps (as presented in the CEBRA demo). We find that CEBRA reaches a performance of more than 90\% in the case of a sample with 40 time steps (Figure 5c in their paper). **After training with a sample of 40 time steps, our model achieves a performance of 94.62±0.48**.
>
> ||200|400|600|800|1000|
> |-|:-:|:-:|:-:|:-:|:-:|
> |**CEBRA**|51.83±0.41|59.92±0.47|64.55±0.70|72.70±0.75|79.90±0.69|
> |**TiDeSPL-VAE**|**54.73±0.35**|**72.40±0.50**|**77.21±0.29**|**79.08±0.30**|**80.59±0.26**|
>
> Table R2: The decoding scores (\%) on the Allen Brain Natural Movie 1 (V1 data) presented in CEBRA.
>
> We further evaluate our model on the [synthetic dataset](https://cebra.ai/docs/demo_notebooks/Demo_synthetic_exp.html) presented in CEBRA. **The results show that our model (92.98±0.02) also outperforms CEBRA (90.60±0.04)**. What's more, although the design of our model is motivated by the properties of visual neural activity, we evaluate it on [motor data](https://cebra.ai/docs/demo_notebooks/Demo_primate_reaching.html) presented in CEBRA. We compared our model with three CEBRA models (CEBRA-Position, CEBRA-Target and CEBRA-Time) which use positional, target and time information as auxiliary variables to select positive and negative samples, respectively. As shown in Table R3, our model outperforms CEBRA-Time, which also uses only time information like ours, and CEBRA-Target, which uses target information, in decoding position. However, our model performs worse in decoding direction.
>
> ||Position|Direction|
> |-|:-:|:-:|
> |**CEBRA-Position**|82.79±0.27|61.01± 0.28|
> |**CEBRA-Target**|51.23±0.05|60.55±0.54|
> |**CEBRA-Time**|50.84±0.36|61.81±0.43|
> |**TiDeSPL-VAE**|62.50±1.64|53.41±0.86|
>
> Table R3: The decoding scores (\%) on the motor data presented in CEBRA.
>
> Decoding motor behavior information with latent variable models has been long and widely explored. However, there are fewer studies using latent variable models to decode visual stimuli, and CEBRA made a great step. Given that the task of the Sensorium Challenge is to predict neural activity using visual stimuli, we believe that our work can make a contribution to the benchmark of the task of decoding visual stimuli with LVMs.
>
> **6. The performance of the separate embeddings.**
>
> In Tables 5 and 8 of the main text, we have extracted the content and style latent representations from the full representations and evaluated their decoding performance respectively. The results show that the content latent representations perform better than the style latent representations, which is consistent with our conceptual interpretation of the two, i.e. that **the content representations are more relevant to visual stimuli**.

---

> ### Author Response · Authors · 2024-11-22
>
> **7. The scientific rationale for the design choice.**
>
> The overall design of our model is not an arbitrary overlay of these advanced technologies, but is scientifically motivated. First of all, we have given a conceptual interpretation to the two latent variables in the case of visual neural activity analysis. As described in lines 64-67, content latent variables correspond to **the component of neural activity driven by the current visual stimulus**, while style latent variables correspond to **the neural dynamics influenced by the internal state of the organism**. Based on the interpretation of the two latent variables, the overall objective is to **construct high-quality low-dimensional latent variables**. Using VAE and the contrastive loss both serve this objective. We clarify each term of our loss function point by point below.
>
> * The reconstruction loss: The basic requirement for high-quality latent variables is to **effectively reconstruct the input spike data**.
>
> * The constrastive loss: As we assume that content latent variables correspond to the component of neural activity driven by the current visual stimuli, we use **self-supervised contrastive learning to make them more relevant to visual stimuli**. Specifically, the positive sample is offset by several time steps from a given sample and both correspond to the same or similar visual stimuli. Consequently, **the content latent variables of the positive pairs are required to be closer, which is achieved by the contrastive loss**. The swap operation helps to further bring the content latent variables of the positive sample pairs closer together.
>
> * The KL divergence: Style latent variables correspond to the neural dynamics influenced by the internal state of the organism, which **contains a lot of intrinsic noise and variability**. Therefore, we build them as random values from a parameterized distribution and apply **a time-dependent prior distribution to guide them**.
>
> **8. Your method is more complicated than those you benchmark, and your ablation studies hint at this as well.**
>
> As discussed in the Responses 4 and 7, the design of our model is complex but has a clear conceptual interpretation and scientific motivation. The number of trainable parameters in our model is slightly higher than some models and lower than CEBRA. In Table 4, the decrease in model performance when conducting ablation studies on the loss function (which is larger only for decoding natural movie in Mouse 2) exactly demonstrates **the validity of our scientifically reasonable design**, rather than showing that complexity drives performance. What's more, in the ablation studies on the model structure, **we have provided results of replacing GRU with vanilla RNN (simple MLP), showing that our model still outperforms baselines in most cases**.

---

> ### Author Response · Authors · 2024-11-25
> **Looking forward to reviewer's feedback**
>
> Dear Reviewer XzNA,
>
> Thank you for your valuable time and constructive comments. As the discussion period will end soon, we hope to discuss whether your concerns have been addressed and would appreciate it if you could reconsider the rating. If you have any further questions, we would be happy to discuss them with you and do our best to address them.
>
> Best,
>
> The authors

---

> ### Author Response · Authors · 2024-11-28
>
> Thanks for your feedback.
>
> In fact, CEBRA only used "offset1-model" in the experiments of synthetic data which has no temporal relationships. For all neural datasets, CEBRA used "offset10-model" or "offset40-model" (as described on page 13 of their paper). As shown in https://github.com/AdaptiveMotorControlLab/CEBRA/blob/main/cebra/models/model.py, the former takes the surrounding 10 time points centered on the current point and the latter takes the surrounding 40 time points. Therefore, we think our description is correct. Besides, CEBRA uses time convolution to encode temporal features, which captures the causal relationships among all time points without considering the chronological relationships. Although time convolution is a good method for extracting temporal relationships, it is indeed less natural compared to our model.

---

> ### Comment · Reviewer_XzNA · 2024-11-28
>
> My final comment is that I re-read the PDF at its current state.
>
> I still believe the introduction and related works need improvement.
>
>  I also still believe benchmarking on the NLB would make a lot of sense and insure fair comparisons.
>
> The  natural movie 1 decoding table doesn’t show cebra-behavior, which likely outperforms your method (and is what those authors used).
>
> The overall novelty of the work is minimal as it’s effectively a time extension to  swapVAE, as other reviewers have pointed out.
>
> I also disagree with your “natural” statements - to expand, you use recurrency (RNN) - you then also use “future and past” information. And people use RNNs as models of the brain, especially for HC and M1 because it’s doing forward planning, meaning neural activity will not be time aligned with many internal brain states, and forward trajectories are often simulated. The HC (hippocampus) literature if full of examples of forward “preplay” and “replay” of simulated future behaviors. Thus, arguing that generalists algorithms like LFADS and CEBRA are unnatural just doesn’t make sense to me and hurts your presentation.
>
> I will not be raising my score. I encourage you to reformulate your presentation, fairly show results, and resubmit. I think there is value in the work, but it’s not well presented in the current form.

---

> ### Author Response · Authors · 2024-11-29
>
> Thanks for your pointed feedback.
>
> 1\. We will improve our introduction and related work.
>
> 2\. Although our work focuses on visual brain regions and the NLB is based on motor brain regions, we will test our model on the NLB in the future. More importantly, we believe that our existing experimental comparisons are fair enough, since for the other benchmark models we have ensured that the training losses converge and their optimal performance is obtained through a grid search of the hyperparameters.
>
> 3\. Indeed, the results we report is CEBRA-Behavior, since we directly follow their [demo](https://cebra.ai/docs/demo_notebooks/Demo_Allen.html).
>
> 4\. We not only novelly introduce time dependence, but also make a new conceptual interpretation for the split latent representations in the case of visual neural activity analysis. For fair comparisons, we have built a small version of our model with fewer trainable parameters than Swap-VAE. The results demonstrate the superiority of our model over Swap-VAE.
>
> 5\. First, although we use RNNs, they do not contain future information, but only past and current information, as shown by all the formulas in our manuscript. Second, we agree that neural activity may not be time aligned with brain states or behaviors. However, in terms of the time dependence within neural activity itself, "forward planning" and "forward preplay" just show that the brain uses past information to predict future information, rather than directly using future information, which is not the same thing. "replay" also uses the information from the past. Therefore, the "natural" statements point at both the time dependence and non-use of future information.
>
> In conclusion, we will improve our manuscript, but we believe that we have made fair and sufficient comparisons and our statements about "naturally incorporating time dependence" are correct.

---

### Official Review · Reviewer_umrd · 2024-11-04

**Soundness:** 2
**Presentation:** 2
**Contribution:** 2
**Rating:** 3
**Confidence:** 4

**Summary:**

A latent variable model is described and used to reanalyze a publicly available dataset of population recordings in mouse visual cortex.
In particular, this work proposes a time series analysis of the neural dynamics in response to videos.
This method combines a variational auto-encoder loss and a self-supervised contrastive learning objective in the latent space.
The latent space is split into a static state and a dynamic state, and both evolve according to learned recurrent dynamics.

**Strengths:**

As large population recordings neural datasets are being measured, developing correspondingly flexible latent time series models is an important research direction.
This work proposes an architecture and unsupervised objective for discovering structure in such neural data.
The focus on neural dynamics in response to video input is timely and relevant.
The presentation begins with simple toy synthetic experiements and then builds to the analysis of a rich neural dataset.

**Weaknesses:**

The proposed architecture is quite complex and no clear justification is provided for the design of factorized latent states. Moreover the latent states are also swapped for the computation of the objective function. I see that these design choices are inherited from a previous publication, but I think that it would help the reader if you could explain clearly the reason for these choices. If a conceptual or mathematical explanation is not possible, ablation experiments could also help (e.g. with vs. without latent state factorization).
The current explanation is vague, e.g. line 231: "To enhance the effect of the positive sample, we adopt the practice of swapping content latent variables between the positive pairs while maintaining style latent variables."

The proposed objective function combines three terms: a VAE style ELBO, a weight decay regularizer, and a contrastive term.
Your reader will find it difficult to think about the overall objective: what is being optimized exactly?
Is this multi-objective optimization still amenable to an interpretation in terms of likelihood?
It seems like you are combining a generative and a disctiminative objective, I wonder if it may be possible to express all that in a nice unified and coherent Bayesian framework, rather than simply saying that you added many terms.
This is also important for the interpretation of the results, it would help clarify what each of these part is contributing and may also help selecting natural hyperparameters.
The writing is vague and should be clarified, for example:
line 74: "we apply self-supervised contrastive learning to enhance the time constraint and to shape latent variables"
line 134: "The output ... is an estimate of firing rates of the input."
line 213: "Besides, we added L2 regularization to the expectation and log-variance of the prior distribution to stabilize model training."

The results are difficult to interpret because you compare methods that are all quite sophisticated.
It would help if, for each task, you reminded your reader where chance performance is.
Another way to do that would be to provide some simple linear method baseline.
Some of the writing is unclear, e.g. line 298: "The dataset contains 32 sessions, each for one mouse. Since the class of neurons responsive to natural visual stimuli is found in six visual regions, in this work we choose to analyze the neural activity of five mice that have as many neurons as possible (about 300, see Appendix D for exact numbers) with them evenly distributed across all regions (the coefficient of variation for the number of neurons across six brain regions is below 0.5)."

Most of the evaluations are concerned with properties of the learned representations.
Adding a discussion of the performance of the autoencoding performance would help.
How well are the spikes reconstructed by the autoencoder, can you show examples of successes and failures?

**Questions:**

Have you considered using only the contrastive objective, i.e. without the decoder and VAE loss? I am asking this because all your benchmarks quantify decodability of image identity from the learned representation.

You write line 303 "We select five scenes that elicit the strongest average responses".
Could you tell us more about the images you selected, and maybe show examples?
How does you method behave on the other images, is the approach robust enough to apply to lower signal to noise examples?

Again for the natural videos, it would help if you could give a precise description of the stimulus.
In particular, how much motion is there in these videos?
This is important because you define a very liberal criterion for correct decoding, line 447: "we take the accuracy measured by considering the error between a predicted frame and the true frame within 1s (default size of time window constraint) as a correct prediction."

In line 409 and following, you interpret the shape of the tSNE visualizations.
Is it safe to interpret such a nonlinear embedding? How much of your interpretation depends on hyperparameter choices?

There is a very large performance difference between Mouse 1 and others in Table 3 and 4, what do you make of the variability and can you help us understand this outlier? You already removed one mouse from your analysis, what was the reasoning behind that?

The original paper from which you get the neural data was focused on describing the mouse visual hierarchy. Can your method reveal interesting structure in each visual area and can it reveal a gradient in the visual representation along the cortical hierarchy?

---

> ### Author Response · Authors · 2024-11-22
>
> We thank the reviewer for the thoughtful and detailed comments. We will do our best to address the comments and answer the questions. Below are our detailed responses.
>
> **1. The justification for the design of factorized latent states.**
>
> The design of two latent states in the previous publication is motivated by specific motor behaviors. In our work, we introduce time dependence into the two latent states by recurrent modules to improve their construction and have given them a new conceptual interpretation in the case of visual neural activity analysis. As described in lines 64-67, content latent states correspond to **the component of neural activity driven by the current visual stimulus**, while style latent states correspond to **the neural dynamics influenced by the internal state of the organism**.
>
> In Tables 5 and 8 of the main text, we have extracted the content and style latent states from the full states and evaluated their decoding performance respectively. In addition, we train two models without each of the latent states as suggested by the reviewer, and report the results in Tables R1 and R2. These results suggest that the factorized latent states are important and that the content latent states perform better than the style latent states, which is consistent with our conceptual interpretation of the two, i.e. that **the content states are more relevant to visual stimuli**.
>
> |Model|Mouse 1|Mouse 2|Mouse 3|Mouse 4|Mouse 5|
> |-|:-:|:-:|:-:|:-:|:-:|
> |**TiDeSPL-VAE**|**96.4±1.1**|**74.8±2.0**|**74.8±1.7**|**78.8±2.9**|**67.6±2.1**|
> |Only with content|74.0±2.0|49.6±2.8|53.2±2.3|68.8±3.8|56.0±3.1|
> |Only with style|60.0±3.0|32.4±2.6|44.4±2.6|47.2±1.7|44.8±2.8|
>
> Table R1: The ablation results (\%) for the latent states in natural scene decoding experiments.
>
> |Model|Mouse 1|Mouse 2|Mouse 3|Mouse 4|Mouse 5|
> |-|:-:|:-:|:-:|:-:|:-:|
> |**TiDeSPL-VAE**|**13.88±0.19**|**65.38±0.36**|**59.88±0.72**|**54.33±0.54**|30.18±0.40|
> |Only with content|12.16±0.24|63.33±0.29|58.64±0.37|52.11±0.39|**30.24±0.54**|
> |Only with style|7.57±0.24|11.73±0.43|12.86±0.35|16.10±0.33|9.89±0.22|
>
> Table R2: The ablation results (\%) for the latent states in natural movie frame decoding experiments.

---

> ### Author Response · Authors · 2024-11-22
>
> **2. The clarifications of objective function.**
>
> The overall objective is to **construct high-quality low-dimensional latent variables based on our interpretation of the two latent states** (see Response 1). The VAE style ELBO derived from the likelihood of spike data and the contrastive loss both serve this objective rather than being considered separately. We clarify each term of our loss function point by point below.
>
> * The reconstruction loss: The basic requirement for high-quality latent variables is to **effectively reconstruct the input spike data**. Since directly reconstructing spike data poses difficulties for model training, it is commonly assumed that **the input data follow a Poisson distribution**, and the output of models is the reconstruction of the firing rate of the Poisson distribution, as described in line 134. This approach and formulation has been widely used in related work [1-3].
>
> * The constrastive loss: As we assume that content latent states correspond to the component of neural activity driven by the current visual stimuli, we use **self-supervised contrastive learning to make them more relevant to visual stimuli**. Specifically, the positive sample is offset by several time steps from a given sample and both correspond to the same or similar visual stimuli. Consequently, **the content latent states of the positive pairs are required to be closer, which is achieved by the contrastive loss**. The offset is less than the length of the sample to ensure that the positive sample pairs overlap, **providing a strong time constraint**. As for the sentence in line 231, we use formulas for clear explanation. For the latent variables of a given sample $z=[z^{(c)},z^{(s)}]$ and the positive sample $z\_{pos}=[z^{(c)}\_{pos},z^{(s)}\_{pos}]$, we swap their content latent states and keep the style latent states, resulting in $\hat{z}=[z^{(c)}\_{pos},z^{(s)}]$ and $\hat{z}\_{pos}=[z^{(c)},z^{(s)}\_{pos}]$. The swapped latent variables are used to compute new reconstructed firing rates $\hat{r}$ and $\hat{r}\_{pos}$, and the additional reconstruction loss $L\_P(x,\hat{r})+L\_P(x_{pos},\hat{r}\_{pos})$, which help to further bring the content latent states of the positive sample pairs closer together.
>
> * The KL divergence: Style latent states correspond to the neural dynamics influenced by the internal state of the organism, which **contains a lot of intrinsic noise and variability**. Therefore, we build them as random values from a parameterized distribution and apply **a time-dependent prior distribution to guide them**. To avoid excessive fluctuations over time, we use the L2 norm of the mean and log-variance of the prior distribution (assumed to be Gaussian) as a regularization loss term.
>
> The ablation results in Section 4.4 demonstrate that these loss terms contribute to the construction of high-quality latent variables. We will include these clarifications of our objective function in the revision and make it more readable.
>
> **3. The results are difficult to interpret because you compare methods that are all quite sophisticated.**
>
> In the experiments on the mouse visual neural dataset, **we have provided the decoding scores of a simple linear method, PCA** (Tables 2 and 3 in the main text). We apply PCA to neural activity and obtain embeddings with the same dimensions as latent variables of network models for the decoding experiments. We have also made comparisons with vanilla VAE ($\beta$-VAE) as a simple network model. In addition, we present the results of PCA on the two synthetic datasets here (88.71±0.01 for the synthetic non-temporal dataset and 20.76±0.08 for the synthetic temporal dataset).
>
> We are sorry for the unclear description of data selection in line 298. In the analysis, we actually selected 5 mice (5 sessions) that have the highest number of recorded neurons, and these recorded neurons are evenly spread across six visual areas. We will revise the description in the revision.

---

> ### Author Response · Authors · 2024-11-22
>
> **4. How well are the spikes reconstructed by the autoencoder?**
>
> Since the output of models is the reconstruction of the firing rate, we compute the root mean square error (RMSE) between the PSTH of the real spike data and the reconstructed firing rate of each model over time as a metric to evaluate the reconstruction performance, which has been used in related work [2-3]. As shown in Tables R3 and R4, our model performs slightly worse than Swap-VAE but better than LFADS and pi-VAE, suggesting that our model is effective in reconstructing neural activity.
>
> |Model|Mouse 1|Mouse 2|Mouse 3|Mouse 4|Mouse 5|
> |-|:-:|:-:|:-:|:-:|:-:|
> |LFADS|0.0471±0.0009|0.0541±0.0012|0.0522±0.0010|0.0500±0.0011|0.0469±0.0009|
> |pi-VAE|0.0548±0.0013|0.0636±0.0017|0.0585±0.0014|0.0604±0.0016|0.0544±0.0012|
> |Swap-VAE|**0.0408±0.0007**|**0.0440±0.0008**|**0.0439±0.0008**|**0.0428±0.0008**|0.0429±0.0008|
> |**TiDeSPL-VAE**|0.0453±0.0009|0.0496±0.0011|0.0474±0.0009|0.0466±0.0010|**0.0424±0.0007**|
>
> Table R3: The reconstruction results (\%) on mouse neural dataset under natural scene stimuli.
>
> |Model|Mouse 1|Mouse 2|Mouse 3|Mouse 4|Mouse 5|
> |-|:-:|:-:|:-:|:-:|:-:|
> |LFADS|0.0557±0.0016|0.0670±0.0022|0.0655±0.0021|0.0678±0.0022|0.0641±0.0021|
> |pi-VAE|0.0559±0.0016|0.0675±0.0022|0.0663±0.0021|0.0686±0.0023|0.0643±0.0021|
> |Swap-VAE|**0.0432±0.0011**|**0.0506±0.0015**|**0.0484±0.0014**|**0.0512±0.0016**|**0.0483±0.0014**|
> |**TiDeSPL-VAE**|0.0466±0.0013|0.0562±0.0018|0.0552±0.0017|0.0563±0.0019|0.0543±0.0018|
>
> Table R4: The reconstruction results (\%) on mouse neural dataset under natural movie stimuli.
>
> **5. Have you considered using only the contrastive objective, i.e. without the decoder and VAE loss?**
>
> As described in Response 2, all the loss terms are based on our conceptual interpretation of the latent variables in relation to visual neural activity, and all play a crucial role in the construction of high-quality latent variables. Indeed, one of the models used for comparison in our work, CEBRA, is the model that uses only the encoder and the contrastive loss and our model outperforms it in almost all experiments.
>
> **6. About the natural scenes.**
>
> The five scenes that elicit the strongest average responses are eagle, lion, cat, bird, and salamander. These are all animals, some of which are natural enemies of mice, so it makes sense that they would elicit the strongest responses. As suggested by the reviewer, we choose five other scenes that elicit the weakest responses. They are forests, withered leaves, mountains, geese, and reeds, which also makes sense. We conduct two experiments, one with these new scenes only and one with all ten scenes. As shown in Tables R5 and R6, **our model performs best in most cases, demonstrating its robustness**.
>
> |Model|Mouse 1|Mouse 2|Mouse 3|Mouse 4|Mouse 5|
> |-|:-:|:-:|:-:|:-:|:-:|
> |LFADS|44.8±3.5|23.2±3.5|26.4±1.4|36.8±3.3|35.2±3.6|
> |pi-VAE|20.0±2.5|28.8±3.1|29.6±2.7|20.8±0.7|19.2±0.7|
> |Swap-VAE|47.2±2.6|32.8±3.1|26.4±0.9|33.6±3.3|28.8±1.3|
> |CEBRA|36.0±3.0|27.2±2.1|29.6±1.4|32.8±1.3|**36.0±1.1**|
> |**TiDeSPL-VAE**|**58.4±2.7**|**35.2±3.5**|**36.8±2.4**|**42.4±2.1**|32.0±1.6|
>
> Table R5: The decoding results (\%) on five new scenes.
>
> |Model|Mouse 1|Mouse 2|Mouse 3|Mouse 4|Mouse 5|
> |-|:-:|:-:|:-:|:-:|:-:|
> |LFADS|49.6±3.2|49.2±3.9|42.0±0.6|47.6±1.3|43.6±2.2|
> |pi-VAE|56.0±2.2|55.6±3.3|**53.6±2.9**|47.6±1.5|37.2±0.9|
> |Swap-VAE|46.4±5.2|38.4±2.1|29.6±1.4|44.8±1.5|38.0±1.3|
> |CEBRA|30.4±1.5|26.4±1.8|27.6±2.5|29.6±2.1|22.4±3.0|
> |**TiDeSPL-VAE**|**68.4±2.0**|**56.8±1.8**|51.6±2.5|**51.6±0.9**|**45.6±1.8**|
>
> Table R6: The decoding results (\%) on all ten scenes.
>
> Compared to the experiments on the scenes that elicit the strongest responses, the performance on the scenes that elicit the weakest responses is lower overall, suggesting that it is more difficult for models to build stimulus-relevant latent variables from neural activity with low signal-to-noise.
>
> We will provide some examples of natural scenes in the revision.
>
> **7. About the natural movie.**
>
> The natural video is a 30-second clip extracted from a movie. Most of the film consists of perspective changes of background objects (streets, buildings, parked cars, etc.) with a small amount of character movement. There are few scenes of fast motion. The criterion for correct decoding follows the previous publication, CEBRA [4], as described in line 431 of our manuscript.
>
> We will provide the natural movie in the supplementary material.
>
> **8. About the tSNE visualizations.**
>
> We try different hyperparameters (*perplexity* in the range [5, 50] and *early_exaggeration* in the range [12, 24]) and the properties explained in our work of embedding are stable. Besides, this visualization method has been used in related work [3-4] and can effectively cluster similar samples. Therefore, we believe that it is safe to interpret these nonlinear embeddings. We will include some visualization with different hyperparameters in the Appendix of the revision.

---

> ### Author Response · Authors · 2024-11-22
>
> **9. About the performance difference.**
>
> The large performance difference may be due to the fact that neural responses elicited by complex natural visual stimuli show large variability across subjects [5]. It is difficult for latent variable models to consistently construct high-quality stimulus-relevant latent representations across conditions and this requires further exploration. We have discussed this limitation in lines 533-537.
>
> Besides, we think you may have a misunderstanding about Table 4. We didn't remove any mice in the ablation studies. The four columns of results are for Mouse 1 and Mouse 2 under natural scenes and natural movie respectively. The results for the remaining three mice are shown in Table 7 in the Appendix.
>
> **10. About the mouse visual hierarchy.**
>
> Our work focuses on **revealing the intrinsic correlation between neural activity and visual stimuli** and puts neurons from all brain regions together as inputs. Therefore, we could not analyze the hierarchy of the mouse visual cortex from the existing results. However, this is a meaningful and constructive suggestion. In the future, we will try to extract latent variables from each brain region separately, and the transformation relationship of latent variables between different brain regions may be able to reveal the mouse visual hierarchy.
>
> [1] Chethan Pandarinath, et al. Inferring single-trial neural population dynamics using sequential auto-encoders. Nature methods, 2018.
>
> [2] Ding Zhou and Xue-Xin Wei. Learning identifiable and interpretable latent models of highdimensional neural activity using pi-vae. NeurIPS, 2020.
>
> [3] Ran Liu, et al. Drop, swap, and generate: A self-supervised approach for generating neural activity. NeurIPS, 2021.
>
> [4] Steffen Schneider, Jin Hwa Lee, and MackenzieWeygandt Mathis. Learnable latent embeddings for joint behavioral and neural analysis. Nature, 2023.
>
> [5] Tyler D Marks and Michael J Goard. Stimulus-dependent representational drift in primary visual cortex. Nature communications, 2021.

---

> ### Author Response · Authors · 2024-11-25
> **Looking forward to reviewer's feedback**
>
> Dear Reviewer umrd,
>
> Thank you for your valuable time and constructive comments. As the discussion period will end soon, we hope to discuss whether your concerns have been addressed and would appreciate it if you could reconsider the rating. If you have any further questions, we would be happy to discuss them with you and do our best to address them.
>
> Best,
>
> The authors

---

> > ### Comment · Reviewer_umrd · 2024-12-02
> >
> > I thank the authors for their responses and additional experiments. However my initial concerns about the soundness of the method remain. I still do not understand the design choices that have lead you to this complex architecture and loss. In your response to my concern, you repeat your conceptual interpretation of i) the latent variables and of their separation into two states, and ii) the combination of a VAE loss with a contrastive SSL loss. These interpretative statements do not constitute an explanation, and although the author's reasoning may be sound, the scientific logic behind this paper remains unclear to me. For example i) the model separates "content" and "state", but both contribute significantly to the decoding, in contradiction with the claim that visually driven activity is contained in the "content" variable; ii) the model reconstruction accuracy is worse than the baseline swapVAE, indicating that the added contrastive loss competes with the VAE loss. Have you evaluated performance for other choices of hyperparameters? I had suggested using only SSL, but you could trade off the two terms and show us how your results change. You have only shown that there exists a set of hyperparameter for which decoding performance is improved. Unless you can give a clear explanation for your choices, I find it difficult to evaluate the method or to say what can be learned from the results.

---

> ### Author Response · Authors · 2024-12-02
>
> Dear Reviewer umrd,
>
> Thank you for your valuable comments and insightful suggestions. We have updated our manuscript to include the clarifications and additional results. As the extended discussion period draws to a close, we hope to discuss whether your concerns have been addressed and would be very grateful if you could reconsider the rating of our work. If you have any further suggestions or comments, we would be delighted to address them.
>
> Best,
>
> The authors

---

> ### Author Response · Authors · 2024-12-03
>
> Thanks for your pointed feedback.
>
> 1\. The results of Tables 5 and 10 in our manuscript show that the performance of content latent representations is significantly higher than that of style latent representations, suggesting that it is the content latent representations that mainly contribute to decoding. The passable decoding performance of style latent representations may be due to the fact that visual stimuli necessarily affect the organism’s internal state. These results do not run counter to our interpretations of the two latent representations. In future work, we will evaluate our model on other visual neural datasets that include behavior data and validate that style latent representations are more relevant to internal states.
>
> 2\. We agree that the contrastive loss that competes with the VAE loss may result in worse reconstruction performance. However, the main goal of our work is to construct high-quality latent representations and our model has achieved it. We will explore how to address the limitation of reconstruction to aid our model in being good at both.
>
> 3\. As we have shown that the model without the contrastive loss (applied to content latent variables) performs worse (Tables 4 and 9), we perform another ablation study of a model without the KL loss (applied to style latent variables). As shown in Tables R1 and R2 (we report results of two mice due to time constraints), the results demonstrate that the KL loss also contributes to decoding performance. We will trade off the two terms and investigate their effect in future work.
>
> |Model|Mouse 1|Mouse 2|
> |-|:-:|:-:|
> |**TiDeSPL-VAE**|**96.4±1.1**|**74.8±2.0**|
> |Without KL loss|93.6±1.3|63.2±2.2|
>
> Table R1: The ablation results (\%) for the latent states in natural scene decoding experiments.
>
> |Model|Mouse 1|Mouse 2|
> |-|:-:|:-:|
> |**TiDeSPL-VAE**|**13.88±0.19**|**65.38±0.36**|
> |Without KL loss|12.40±0.23|63.23±0.33|
>
> Table R2: The ablation results (\%) for the latent states in natural movie frame decoding experiments.

---

### Author Response · Authors · 2024-12-01

We sincerely thank all reviewers for valuable time and thoughtful and constructive comments. We have done our best to address the concerns raised by reviewers in each individual comment. As the extended discussion period will end soon, **we would like to emphasize our main clarifications and additional results, which we have included in our revised manuscript**.

1. We have provided a more systematic conceptual interpretation and a more detailed description of the design of our model and loss function (as presented in comments to Reviewers umrd, 35tP and MX7e).

2. We have added a more fair comparison with the most competitive model, Swap-VAE. **We build a small version of our model (TiDeSPL-VAE-small) with fewer trainable parameters than Swap-VAE and show that it still outperforms Swap-VAE**, which suggests that the performance improvement mainly stems from the design of our model rather than the larger number of parameters (as presented in comments to Reviewer 35tP).

3. We have added further ablation studies suggested by the reviewers (as presented in comments to Reviewers umrd and 35tP).

4. **We have evaluated our model in other benchmarks** and the results demonstrate the effectiveness of our model (as presented in comments to Reviewer XzNA).

**We would be very grateful if the reviewers could reconsider the rating of our work**.

---

### Meta-Review · Area_Chair_qUiW · 2024-12-20

**Metareview:**

The paper introduces TiDeSPL-VAE, a recurrent latent variable which combines a variational auto-encoder loss and a self-supervised contrastive learning objective in the latent space. The latent space is split into a static state and a dynamic state, and both evolve according to learned recurrent dynamics.
This model is used to reanalyze a publicly available dataset of population recordings of mice's neural activity during passive viewing of natural images and movies.
The authors show that the TiDeSPL-VAE outperforms numerous baselines in reconstructing spiking activity on synthetic and naturalistic datasets.

Strengths: Reviewers agree that the focus on neural dynamics in response to video input is timely and relevant.
Weaknesses: The architecture is complex and the design choices are not sufficiently motivated. The paper in its current form lacks clarity.
There is substantial disagreement whether the presented experiments are adequate. I.e if they include the relevant baselines and whether or not they support the claims of the paper.

After re-reading the revised paper, reviewer XzNA recommends to "reformulate your presentation, fairly show results, and resubmit" and I agree with that assessment. This work addresses and interesting question and obviously has value, but it is not sufficiently clear in its motivation, presentation and evaluation.

**Additional Comments On Reviewer Discussion:**

Most of the discussion has focussed on 1) the motivationZ/justification of the complexity of the model and 2) on the adequacy of the benchmarks.
During the discussion the authors have provided lots of additional results which addressed some of the raised concerns.
The two most critical reviewers remain unconvinced by this evidence, while 35tP has slightly raised their score.

---

### Decision · Program_Chairs · 2025-01-22

Reject